# Weakly Supervised Object Segmentation by Background Conditional Divergence

**Hassan Baker**                                           *bakerh@udel.edu*
*Department of Electrical and Computer Engineering*
*University of Delaware*
*Newark, DE, 19716 USA*

**Matthew S. Emigh**                         *matthew.s.emigh.civ@us.navy.mil*
*Naval Surface Warfare Center Panama City Division*
*Panama City, FL, 32407 USA*

**Austin J. Brockmeier**                                   *ajbrock@udel.edu*
*Department of Electrical and Computer Engineering, Department of Computer and Information Sciences*
*University of Delaware*
*Newark, DE, 19716 USA*

**Reviewed on OpenReview:** *https://openreview.net/forum?id=2JJZhfGvMW*

## Abstract

As a computer vision task, automatic object segmentation remains challenging in specialized image domains without massive labeled data, such as synthetic aperture sonar images, remote sensing, biomedical imaging, etc. In any domain, obtaining pixel-wise segmentation masks is expensive. In this work, we propose a method for training a masking network to perform binary object segmentation using weak supervision in the form of image-wise presence or absence of an object of interest, which provides less information but may be obtained more quickly from manual or automatic labeling. A key step in our method is that the segmented objects can be placed into background-only images to create realistic images of the objects with counterfactual backgrounds. To create a contrast between the original and counterfactual background images, we propose to first cluster the background-only images and then, during learning, create counterfactual images that blend objects segmented from their original source backgrounds to backgrounds chosen from a targeted cluster. One term in the training loss is the divergence between these counterfactual images and the real object images with backgrounds of the target cluster. The other term is a supervised loss for background-only images. While an adversarial critic could provide the divergence, we use sample-based divergences. We conduct experiments on side-scan and synthetic aperture sonar in which our approach succeeds compared to previous unsupervised segmentation baselines that were only tested on natural images. Furthermore, to show generality we extend our experiments to natural images, obtaining reasonable performance with our method that avoids pretrained networks, generative networks, and adversarial critics. The code for this work can be found at https://github.com/bakerhassan/WSOS.

## 1 Introduction

Automatic foreground object segmentation is a computer vision task with a variety of applications in many fields such as medicine, autonomous driving, and defense (Ma et al., 2024; Song et al., 2024; Lee et al., 2024; Cao et al., 2024; Gao et al., 2024; Marinov et al., 2024; Kamtam et al., 2025). Supervised machine learning approaches are limited by the availability of supervised training annotations, such as object masks

or bounding boxes, which in specialized domains are expensive to obtain (Russakovsky et al., 2015; Bakas et al., 2017; Sohn et al., 2020). Weak supervision in the form of image-level labels is often sufficient to guide segmentation. In particular, a well-trained classifier can provide class-activation maps to guide segmentation (Kumar Singh & Jae Lee, 2017; Wei et al., 2017; Zhang et al., 2018; Li et al., 2018; Xu et al., 2022; Xie et al., 2022). Prior work on unsupervised learning for object segmentation builds off either pretrained classifiers or the underlying feature extraction backbone networks (Remez et al., 2018; Li et al., 2023; Dosovitskiy et al., 2021; Hamilton et al., 2022; Zhang et al., 2018; Xie et al., 2022). However, for specialized imaging domains, such as sonar and medical images, there are not large crowd-sourced labeled datasets, and the limited datasets that *are* available differ in sensors and contexts that make it difficult to pretrain useful models. Thus, we consider the weakly supervised task for which no pretrained networks are available, where the dataset consists of two sets of images: one labeled as having an object (i.e., composite images) and the other as having no object (i.e., background-only images), which can be obtained through a combination of expert annotation, supervised image-level classification, and human-in-the-loop active learning.

Our goal is to train a neural network to generate binary pixel-wise segmentation masks that separate salient foreground objects from the background. A motivating possibility for such a network is the ability to generate synthetic images composed of an object extracted from one image and background pixels from another image (Ostyakov et al., 2018). This synthetic image, created by alpha-blending a composite image and a background-only image, is referred to as a counterfactual image, as the background is counterfactual to the original background. While the automatic creation of counterfactual images is independently useful for augmenting training datasets to resolve spurious correlations between objects and their background, these counterfactual images are key to our approach as the network is trained to minimize the statistical divergence between the distributions of real images and the counterfactual images.[1] To minimize this loss, the learned masks must generate counterfactual images that are statistically similar to the real images. However, without further specification this loss is minimized by a degenerate masking that segments the entire image as the object such that after alpha-blending the output is the original input images. Other work on unsupervised segmentation, which lacks the additional set of background-only images, has used the generators and discriminators underlying generative adversarial networks (GANs) along with additional loss terms to avoid such degenerate solutions (Bielski & Favaro, 2019; Sauer & Geiger, 2021; Zou et al., 2023; Yang et al., 2022b). Another solution proposed by Ostyakov et al. (2018) is to use a generative network to inpaint missing portions of the background after applying the complement mask, which should then be background only, and then calculate the divergence between inpainted images and real background-images.

In contrast, our novel approach to avoid the degenerate solution is relatively straightforward: firstly, we organize all images into clusters based on their background content; secondly, we form counterfactuals with background-only images taken from a target cluster that differs from original background cluster; and finally, we compute the conditional divergence between these counterfactual and real images. Although there are multiple ways to achieve background clustering, we choose to perform K-means clustering (or a modified version that seeks balanced clusters) in a latent space obtained from training either an autoencoder (AE) with reconstruction loss or an encoder with self-supervised contrastive learning on the background-only images. For datasets where foreground objects are roughly centered, the background content can be inferred from random crops near edges and corners. In cases where composite images with objects are patches taken from a larger image, the background is inferred from nearby patches without objects. The approach is summarized in Figure 1. Using the average of the conditional divergences across all clusters as a loss encourages the masking network to learn meaningful segmentations by leveraging background variations while effectively isolating foreground objects.

The differences between our approach and the majority of the other work reviewed in Section 2 are the following:

---

[1]We assume the object can be placed on an independently chosen background. This assumption is valid in underwater sonar where a man-made object's location is largely independent of the background. However, in natural imagery there is much stronger dependence between the object and location within the image, due to depth and context, making it imperative to choose an appropriate background image and/or location within the background, a problem addressed in other work (Dvornik et al., 2018; Zhu et al., 2023).

1. We assume access to two types of images: composite images (i.e., images that contain foreground objects on a background) and background-only images. *These are the only labels that we assume to be available.*

2. The model consists of a masking network whose output is used to alpha-blend the input composite image with a background-only images, producing a synthetic, counterfactual image.

3. During training, we compute the conditional statistical divergence between background-conditioned counterfactual images and real composite images, and we compute a supervised cost for background-only images.

4. Our approach is relatively straightforward; it consists of a single masking network with a composite loss that is a sum of the two terms, without extensive hyperparameters.

5. We do not rely on pretrained models, affording our research the benefit of achieving satisfactory object segmentation results even with limited datasets of modalities, for which pretrained models are unavailable.

In particular, Table 6 shows that the related work suffers from requiring at least one of the following undesired attributes: pretrained models, adversarial training (which is known for its instability), labeled data (e.g., class information), or extensive hyperparameter search for a complicated loss function combination (i.e., not minimalist).

We evaluate the effectiveness of our segmentation approach on the following datasets:

1. **Toy datasets** created by combining canonical object datasets (e.g., MNIST, Fashion MNIST (Xiao et al., 2017), and dSprites (Matthey et al., 2017)) with textures as backgrounds (Brodatz, 1966). We show that existing unsupervised segmentation benchmarks (Bielski & Favaro, 2019; Xie et al., 2022) fail to converge to an acceptable segmentation on these grayscale image datasets.

2. **AI4Shipwrecks** consists of side-scan sonar images of shipwrecks with other foreground objects such as large rocks (Sethuraman et al., 2024). We show that our methodology provides finer-grained details for foreground objects compared to the available ground truth.

3. **SAS-Clutter** consists of small synthetic aperture sonar (SAS) image snippets containing seafloor clutter objects such as crab traps, barrels, tires, etc. Results demonstrate meaningful segmentation. The dataset is not publicly available.

4. **SAS+dSprites** consists of real SAS images of various seafloor types (Cobb & Zare, 2014) with artificial shapes from dSprites (Matthey et al., 2017) as synthetic objects.

5. **CUB-200-2011** consists of natural images of 200 species of birds (Wah et al., 2011).

## 2 Related Work

Our weakly supervised foreground object segmentation is most closely related to prior work on unsupervised foreground object segmentation (Bielski & Favaro, 2019; Sauer & Geiger, 2021; Zou et al., 2023; Yang et al., 2022b). These methods either depend on adversarial training (Ostyakov et al., 2018; Remez et al., 2018; Bielski & Favaro, 2019; Chen & Konukoglu, 2018; Singh et al., 2019; He et al., 2022; Yang et al., 2022a; Zou et al., 2023; Sauer & Geiger, 2021; Li et al., 2023), make assumptions about the foreground object size (Bielski & Favaro, 2019; He et al., 2022; Yang et al., 2022b; Zou et al., 2023; Sauer & Geiger, 2021), depend on a given initial mask (Li et al., 2023), need an extensive hyperparameters search (Li et al., 2018; Remez et al., 2018; Bielski & Favaro, 2019; Singh et al., 2019; He et al., 2022; Yang et al., 2022b; Zou et al., 2023; Sauer & Geiger, 2021; Hung et al., 2019; Savarese et al., 2021), depend on pretrained models or foreground labels (e.g., birds classes) (Remez et al., 2018; Sauer & Geiger, 2021; Li et al., 2023; Dosovitskiy et al., 2021; Hamilton et al., 2022; Wang et al., 2023b; Zhang et al., 2018; Li et al., 2018; Hou et al., 2018; Jiang et al., 2019; Chang et al., 2020; Lee et al., 2021; Wu et al., 2021; Xie et al., 2022; Hung et al., 2019), or require stochastic

generation or complicated combinations of losses to avoid degenerate solutions (Ostyakov et al., 2018; Li et al., 2018; Remez et al., 2018; Bielski & Favaro, 2019; Hung et al., 2019; Zou et al., 2023; Yang et al., 2022b). Our proposed method, on the other hand, does not depend on any of these undesirable attributes. The difference between these prior methods and our work is that we assume access to background-only images that follow the same distribution of backgrounds in the composite images. (Ostyakov et al. (2018) also use this assumption but have an adversarial discriminator along with an enhancement and inpainting network.) While purely unsupervised learning without any background-only images is even more challenging, in practice background-only images can be obtained from the surrounding of images with an object and are naturally obtained through the weak labeling process.

Furthermore, the majority of related work has focused on natural images where there is often visible differences between objects and background due to the optics. Additionally, in other imaging domains, there may also be a lack of pretrained models. In the case of underwater sonar imagery, which is a specialized and limited domain—even more so than standardized medical imaging (Ma et al., 2024)—datasets are limited by the diversity of objects and sensors (Valdenegro-Toro et al., 2021; Preciado-Grijalva et al., 2022). Nonetheless, self-supervised pretraining before classification is possible (Preciado-Grijalva et al., 2022). We found that the majority of the work on unsupervised segmentation of sonar imagery is concerned with sea floor segmentation (Sun et al., 2022; Abu & Diamant, 2017; Yao et al., 2000) or object detection (Abu & Diamant, 2018; Wei et al., 2024). One recent work did investigate generating new sonar imagery using a mask-conditioned diffusion model (i.e., counterfactual images); however, they did not explicitly address segmentation (Yang et al., 2024).

A description of related work is included in Appendix A and summarized in Table 6, which notes which works depend on pretrained models, GAN training, or data labels (e.g., foreground labels). The table highlights that all prior work has at least one of these undesirable criteria, which makes it hard to apply any of these methods to data-scarce domains such as sonar images.

## 3 Methodology

Our methodology trains a masking network by using its output to alpha-blend the input composite image with an independent background-only image creating a counterfactual image. The training loss consists of two terms; the first is a measure of the statistical divergence between counterfactual and real composite images, and the second is a pixel-wise supervised loss that encourages the alpha-blended output to match the input for background-only images. To avoid degenerate solutions we rely on conditioning the divergence based on different background clusters, which we derive from an auto-encoder or contrastive-learning based representation. Additionally, we consider an extra loss for the masks of composite images that forces a portion of the mask to not be background, which is applicable if the object size is known a priori. Our approach requires minimal hyperparameter tuning, namely the number of clusters, epochs, and batch size. For architectures we use a U-Net (Ronneberger et al., 2015b), which is a standard neural network architecture, as the masking network and minimal auto-encoder or encoder-only architectures for the latent representation of backgrounds. Figure 1 summarizes the approach, and Table 1 details the variables in the formulation.

### 3.1 Problem Formulation

We assume two distributions of images: composite images that contain objects surrounded by background and background-only images that do not have salient objects. Let $\mathcal{I} \subset \mathbb{R}^{m \times n \times c}$ denote the space of images with $c$ channels and $m \times n$ pixels, $\mathcal{X} \subset \mathcal{I}$ denote the space of composite images, and $\mathcal{R} \subset \mathcal{I}$ denote the space of background images. Similar to the generative assumptions in the layered GAN (Bielski & Favaro, 2019; Yang et al., 2022b; Zou et al., 2023), we assume a composite image $X_R \in \mathcal{X}$ generated according to

$$X_R = M \circ F + (\mathbf{1} - M) \circ R, \tag{1}$$

where $\circ$ denotes an elementwise product with broadcasting across channels, $R \in \mathcal{R}$ is a background-only image, $F \in \mathcal{I}$ is an image of an object paired with a binary mask $M \in \{0,1\}^{m \times n}$ ($M_{ij} = 1$ if the pixel corresponds to the object and $M_{ij} = 0$ if the pixel corresponds to background), and $\mathbf{1}$ is a $m \times n$-array of

Table 1: Summary of random variable notation used in the problem formulation.

| Notation | Description |
|---|---|
| $(F, M, R) \sim \mathbb{P}_{M,F,R}$ | An object image $F \in \mathcal{I}$, a binary mask $M \in \{0,1\}^{m \times n}$, and a background image $R \in \mathcal{R} \subset \mathcal{I}$ jointly sampled. |
| $R \sim \mathbb{P}_R$ | A background image sampled from the background distribution. |
| $X_R \sim \mathbb{P}_X$ | A composite image constructed from $F$, $M$, and background $R$. |
| $X_{R'}$ | An ideal counterfactual composite image constructed from $F$ and $M$ (or equivalently $X_R$ and $M$) but with background $R' \sim \mathbb{P}_R$. If object-background independence holds $X_{R'} \sim \mathbb{P}_X$, then $X_{R'} \stackrel{d}{=} X_R$, where $\stackrel{d}{=}$ denotes equality in distribution. |
| $M_\theta$ | Masking network $M_\theta : \mathcal{I} \to [0,1]^{m \times n}$. |
| $\tilde{X}_{R'} \sim \mathbb{P}_{\tilde{X}_\theta}$ | A counterfactual image produced from $X_R$ and $R'$ using the mask $M_\theta(X_R)$. Ideally, $\tilde{X}_{R'} \stackrel{d}{=} X_{R'} \stackrel{d}{=} X_R$, assuming object-background independence. |
| $M_{\text{degen}}$ | A degenerate masking function that maps any composite image to a mask of all ones, i.e., $M_{\text{degen}}(X) = \mathbf{1} \in \{1\}^{m \times n}, \quad \forall X \in \mathcal{X}$. |
| $C_b \sim \text{Cat}(1, \dots, K)$ | A random variable for the background cluster label, drawn from a categorical distribution over $K$ clusters. |
| $X_c \sim \mathbb{P}_{X \mid C_b = c}$ | A composite image with a background from cluster $c$. |
| $R_c \sim \mathbb{P}_{R \mid C_b = c}$ | A background image from cluster $c$. |
| $X_{\neg c} \sim \mathbb{P}_{X \mid C_b \neq c}$ | A composite image with a background *not* from cluster $c$. |
| $\tilde{X}_c \sim \mathbb{P}_{\tilde{X}_{c,\theta}}$ | Counterfactual image produced from $X_{\neg c}$ and $R_c$ using the mask $M_\theta(X_{\neg c})$. Ideally, $\tilde{X}_c \stackrel{d}{=} X_c$. |

ones. The unsupervised segmentation goal is to learn to infer $M$ given only $X_{Rc} \sim \mathbb{P}_X$, where $\mathbb{P}_X$ is the distribution of composite images.

Probabilistically, as a random variable $X_R \sim \mathbb{P}_X$ is a function of the dependent random variables $(F, M, R) \sim \mathbb{P}_{M,F,R}$, where the object and mask are dependent, and the object and the background may be dependent. However, we assume $R \sim \mathbb{P}_R$ is independent of $(F, M) \sim \mathbb{P}_{M,F}$. Under this assumption, one can generate a new composite image $X_{R'}$ containing the same object $M \circ X_R = M \circ F$ but with a different independent background $R' \sim \mathbb{P}_R$:

$$X_{R'} = M \circ X_R + (\mathbf{1} - M) \circ R'. \tag{2}$$

While $X_R$ and $R'$ represent real images, $X_{R'}$ is an ideal counterfactual. If the independence assumptions hold, $X_{R'} \sim \mathbb{P}_X$. [2]

A masking network is a function $M_\theta : \mathcal{I} \to [0,1]^{m \times n}$, parameterized by $\theta$, that models the relationship between $X_R$ and $M$.[3] Unlike supervised segmentation, the latter is never observed. An ideal masking network would yield the true mask $M_{\theta^*}(X_R) = M$ when applied to a composite image and would yield an all-zeros mask $\mathbf{0}$ when applied to a background only image $M_{\theta^*}(R) = \mathbf{0}$. The output mask is used to alpha-blend a composite image $X_R \sim \mathbb{P}_X$ with an independent background-only image $R' \sim \mathbb{P}_R$,

$$\tilde{X}_{R'} = M_\theta(X_R) \circ X_R + (\mathbf{1} - M_\theta(X_R)) \circ R' \sim \mathbb{P}_{\tilde{X}_\theta}. \tag{3}$$

The goal is then to make the counterfactual distribution $\mathbb{P}_{\tilde{X}_\theta}$ statistically indistinguishable from the composite image distribution $\mathbb{P}_X$. However, directly minimizing a statistical divergence between $\mathbb{P}_{\tilde{X}}$ and $\mathbb{P}_X$ leads to a trivial solution $M_{\text{degen}} : X \mapsto \mathbf{1}, \quad \forall X \in \mathcal{X}$, where the masking network always produces a mask of all ones (indicating all foreground) so that each generated counterfactual image is the same as its input composite image. That is,

$$M_{\text{degen}} = \underset{M_\theta}{\arg\min}\, D(\mathbb{P}_{\tilde{X}_\theta}, \mathbb{P}_X), \tag{4}$$

---

[2]In domains with strong dependence between objects and background, in order to create the counterfactual $X_{R'}$, the background $R'$ would need to be conditional on the object $F, M$, a problem considered in other works (Fang et al., 2019; Dvornik et al., 2018; Zhu et al., 2023).

[3]While an ideal masking network would have binary output, to facilitate training the unit interval is allowed.

based on the fact that $\tilde{X}_{R'} = M_{\text{degen}}(X_R) \circ X_R + (1 - M_{\text{degen}}(X_R)) \circ R' = X_R \sim \mathbb{P}_X$, a problem also encountered and addressed in different ways in GAN-based approaches for unsupervised segmentation (Bielski & Favaro, 2019; Yang et al., 2022b; Zou et al., 2023) and weakly supervised segmentation (Ostyakov et al., 2018).

To overcome this, we create a distributional shift between the random backgrounds $R$ found in the composite images and the random background-only images. We restrict the counterfactuals $\tilde{X}_{R'}$ to be generated from backgrounds $R'$ that are different from the backgrounds $R$ of the composite images $X_R$. The divergence to be minimized is then computed only between distributions of the same background. This discourages masks from retaining the original (composite) backgrounds $R$, as this would increase the (background-conditional) divergence. Specifically, we assume that all backgrounds belong to a latent background class $C_b \in \{1, \ldots, K\}$, and that there exists a joint distribution $(R, C_b) \sim \mathbb{P}_{R,C_b}$ over background images and latent background class. Let $\mathbb{P}_{X|C_b=c}$ represent the distribution of composite images generated under the condition that the background image belongs to class $c \in \{1, \ldots, K\}$. To target composite images with a specific background class $\mathbb{P}_{X|C_b=c}$, the goal is to generate counterfactuals from independent background-only images of the same class $R_c \sim \mathbb{P}_{R|C_b=c}$ and composite images that do not have this class $X_{\neg c} = X_{R|C_b \neq c} \sim \mathbb{P}_{X|C_b \neq c}$, where $\mathbb{P}_{R|C_b \neq c}$ is the distribution over other background classes, such that $X_c \sim \mathbb{P}_{X|C_b=c}$ and $X_{\neg c}$ are independent random variables with different distributions. The masking network's counterfactual image with background from class $c$ is

$$\tilde{X}_c = M_\theta(X_{\neg c}) \circ X_{\neg c} + (1 - M_\theta(X_{\neg c})) \circ R_c \sim \mathbb{P}_{\tilde{X}_{c,\theta}}. \tag{5}$$

Figure 1 shows a diagram of the counterfactual background-conditional image generation process.

We optimize the parameters $\theta$ of the masking network $M_\theta$ in terms of a background conditional divergence between the distributions of $\mathbb{P}_{X|C_b=c}$ and $\mathbb{P}_{\tilde{X}_{c,\theta}}$,

$$\mathcal{L}_D(\theta) = \sum_c D(\mathbb{P}_{\tilde{X}_{c,\theta}}, \ \mathbb{P}_{X|C_b=c}). \tag{6}$$

Due to the distinct distribution of $X_{\neg c}$ and $X_c$, the trivial masking solution will not minimize the divergence between $\mathbb{P}_{\tilde{X}_{c,\theta}}$ and $\mathbb{P}_{X|C_b=c}$, $M_{\text{degen}}(X_{\neg c}) \circ X_{\neg c} + (1 - M_{\text{degen}}(X_{\neg c})) \circ R_c = X_{\neg c}$ is clearly not distributed according to $\mathbb{P}_{X|C_b=c}$. We summarize the notation in Table 1.

Since we assume a sample of background-only images, we also train the masking network to determine if there is no object in the input image (by outputting an all-zero mask). While a per-pixel binary cross-entropy loss could be used, we use the squared error loss

$$\mathcal{L}_{\text{Bg}}(\theta) = \mathbb{E}_R \Big[ \sum_{ij} [M_\theta(R)]_{ij}^2 \Big] = \mathbb{E}_R \left[ \|M_\theta(R)\|_F^2 \right]. \tag{7}$$

The optimization problem for training the masking network is then

$$\min_{\theta \in \Theta} \mathcal{L}_D(\theta) + \mathcal{L}_{\text{Bg}}(\theta), \tag{8}$$

where $\Theta$ is the set of possible parameters for the masking network.

In practice, the training data are $\{(x_i, c_i)\}_{i=1}^{N_{\text{Comp.}}} \subset \mathcal{X} \times \{1, \ldots, K\}$ and $\{(r_i, c_i')\}_{i=1}^{N_{\text{Bg}}} \subset \mathcal{R} \times \{1, \ldots, K\}$, where $N_{\text{Comp.}}$ and $N_{\text{Bg}}$ are the total number of composite and background-only images, respectively. We define the empirical conditional distributions $\mathbb{P}_{X|C_b=c} = \frac{1}{|\{i|c_i=c\}|} \sum_{i|c_i=c} \delta_{x_i}$, $\mathbb{P}_{X|C_b \neq c} = \frac{1}{|\{i|c_i \neq c\}|} \sum_{i|c_i \neq c} \delta_{x_i}$, and $\mathbb{P}_{R|C_b=c} = \frac{1}{|\{i|c_i'=c\}|} \sum_{i|c_i'=c} \delta_{r_i}$, where $\delta_x$ is the Dirac measure $\delta_x(\mathcal{A}) = \begin{cases} 1 & x \in \mathcal{A} \\ 0 & x \notin \mathcal{A} \end{cases}$, $\forall \mathcal{A} \subseteq \mathcal{I}$. For a given background $c$, three sets of mini-batches of size $B$ are sampled $\{x_{\neg c}^{(i)}\}_{i=1}^B \overset{\text{i.i.d.}}{\sim} \mathbb{P}_{X|C_b \neq c}$, $\{r_c^{(i)}\}_{i=1}^B \overset{\text{i.i.d.}}{\sim} \mathbb{P}_{R|C_b=c}$, $\{x_c^{(i)}\}_{i=1}^B \overset{\text{i.i.d.}}{\sim} \mathbb{P}_{X|C_b=c}$, which correspond to $X_c$, $X_{\neg c}$, and $R_c$, respectively. Then the counterfactuals are produced $\tilde{x}_c^{(i)} = (1 - M_\theta(x_{\neg c}^{(i)})) \circ r_c^{(i)} + M_\theta(x_{\neg c}^{(i)}) \circ x_{\neg c}^{(i)}$, $i \in \{1, \ldots, B\}$. Finally, the divergence $D(\hat{\mathbb{P}}_{\tilde{X}_{c,\theta}}, \hat{\mathbb{P}}_{X|C_b=c})$ is computed from the empirical distributions $\hat{\mathbb{P}}_{\tilde{X}_{c,\theta}} = \frac{1}{B} \sum_{i=1}^B \delta_{\tilde{x}_c^{(i)}}$ and $\hat{\mathbb{P}}_{X|C_b=c} = \frac{1}{B} \sum_{i=1}^B \delta_{x_c^{(i)}}$. Denoting the mini-batch loss functions as $\hat{\mathcal{L}}$, we summarize the training procedure in Algorithm 1.

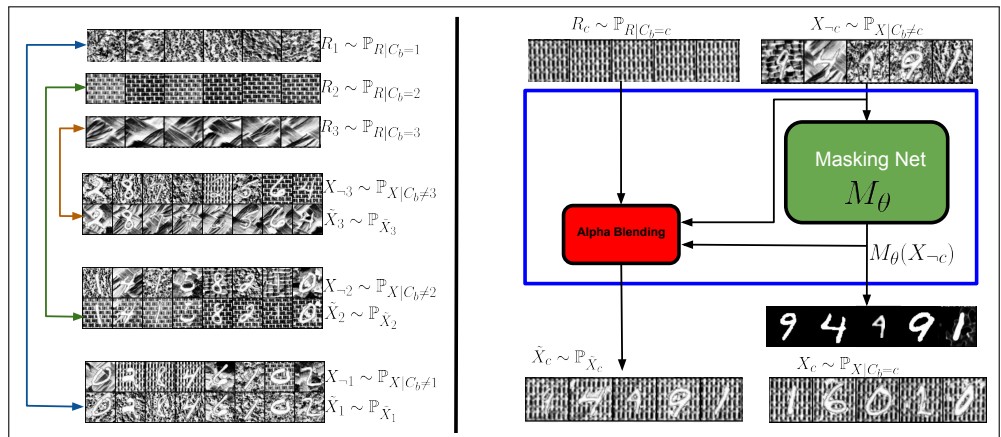

Figure 1: Right: The counterfactual generation model employs a masking network, $M_\theta$, to process composite images, $(X_{\neg c})$, which contain backgrounds other than $c$. These composite images are combined with corresponding background images, $R_c$, of background $c$. The model generates a mask to separate the object from the background, and alpha-blending (as per equation 1) is used to combine the composite images and background images. This results in a counterfactual image that retains the foreground object from $X_{\neg c}$ but places it against a different background. In this instance, the counterfactual image preserves the foreground object in $X_{\neg c}$ but is presented against a distinct background. Left: Demonstration of the inputs and the outputs of the proposed framework indicated by the blue box on the right. The first three rows display different sets of background images, where each set belongs to the same background label. Each subsequent pair of rows shows composite images with various backgrounds, excluding the target background (indicated by the two headed arrow). The second row in each pair presents the generated counterfactual images, where the foreground object is pasted onto the target background using the model on the right.

---

**Algorithm 1** Training Procedure

---

**Require:** Model $M_\theta$, distributions $\mathbb{P}_{X|C_b=c}$, $\mathbb{P}_{X|C_b\neq c}$, and $\mathbb{P}_{R|C_b=c}$ for $c \in \{1, \ldots, K\}$, batch size $B$

1: **for** each training iteration **do**
2:     **for** each category $c \in \{1, \ldots, K\}$ **do**
3:         Sample mini-batches: $\{x_{\neg c}^{(i)}\}_{i=1}^B \overset{\text{i.i.d.}}{\sim} \mathbb{P}_{X|C_b\neq c}$,    $\{r_c^{(i)}\}_{i=1}^B \overset{\text{i.i.d.}}{\sim} \mathbb{P}_{R|C_b=c}$,    $\{x_c^{(i)}\}_{i=1}^B \overset{\text{i.i.d.}}{\sim} \mathbb{P}_{X|C_b=c}$.
4:         $\tilde{x}_c^{(i)} = (1 - M_\theta(x_{\neg c}^{(i)})) \circ r_c^{(i)} + M_\theta(x_{\neg c}^{(i)}) \circ x_{\neg c}^{(i)}, \quad i \in \{1, \ldots, B\}$
5:         $\hat{\mathcal{L}}_D^{(c)}(\theta) = D(\hat{\mathbb{P}}_{\tilde{X}_{c,\theta}}, \hat{\mathbb{P}}_{X|C_b=c}), \quad \hat{\mathbb{P}}_{\tilde{X}_{c,\theta}} = \frac{1}{B}\sum_{i=1}^B \delta_{\tilde{x}_c^{(i)}}, \quad \hat{\mathbb{P}}_{X|C_b=c} = \frac{1}{B}\sum_{i=1}^B \delta_{x_c^{(i)}}$.
6:         $\hat{\mathcal{L}}_{\text{Bg}}^{(c)}(\theta) = \frac{1}{B}\sum_{i=1}^B \|M_\theta(r_c^{(i)})\|_F^2$ (Background loss)
7:     **end for**
8:     Update $\theta$ using gradient step on the total loss: $\hat{\mathcal{L}}(\theta) = \sum_{c=1}^K \left(\hat{\mathcal{L}}_D^{(c)}(\theta) + \hat{\mathcal{L}}_{\text{Bg}}^{(c)}(\theta)\right) = \hat{\mathcal{L}}_D(\theta) + \hat{\mathcal{L}}_{\text{Bg}}(\theta)$.
9: **end for**

---

## 3.2 Statistical Divergence Choice

The choice of the divergence $D$ will necessarily impact performance, computational cost, and the learning dynamics. To connect our approach to supervised segmentation tasks, the Wasserstein-2 distance is natural as it seeks a joint distribution $\mathbb{P}_{\tilde{X}_{c,\theta} X_c}$ that pairs the counterfactual $\tilde{X}_c$ and composite image $X_c$ to be as close as possible in a root-mean squared error sense

$$W_2(\mathbb{P}_{\tilde{X}_{c,\theta}}, \mathbb{P}_{X|C_b=c}) = \inf_{\mathbb{P}_{\tilde{X}_{c,\theta} X_c} \in \mathcal{P}} \sqrt{\mathbb{E}_{(\tilde{X}_c, X_c) \sim \mathbb{P}_{\tilde{X}_{c,\theta} X_c}} \|\tilde{X}_c - X_c\|_2^2}, \tag{9}$$

where $\mathcal{P}$ is the set of joint distributions with marginals equal to $\mathbb{P}_{\tilde{X}_{c,\theta}}$ and $\mathbb{P}_{X|C_b=c}$. The ideal joint distribution would pair the object in $X_c$ with a corresponding object in $X_{\neg c}$ and the background of $X_c$ with a corresponding background $R_c$ such that $\tilde{X}_c$ would match $X_c$ if the masking is correct. However, if the mask-

ing network is wrong, the optimal joint distribution may create non-ideal pairings to compensate, possibly leading to local minima.

Estimating the Wasserstein distance in this primal form is challenging. Based on a review of divergences, including alternatives for the Wasserstein distance, in Appendix B, we select $D$ to be the energy-based sliced Wasserstein (EBSW) distance (Nguyen & Ho, 2023), which avoids adversarial training of discriminators required for the variational estimates of $f$-divergences in GANs (Goodfellow et al., 2014; Nowozin et al., 2016) and has desirable computational and statistical characteristics. In particular it does not suffer from the curse of dimensionality (Nguyen & Ho, 2023), making it broadly applicable without many hyperparameters. For the hyperparameters of the estimate of the EBSW, described by equation 14 in Appendix B, we choose $p = 2$ to match equation 9, $f$ to be the exponential function, and $L = 1000$ slices.

### 3.3 Background Clustering

Generally, we do not assume access to the background class/cluster, although in certain situations this side information is known. We propose to use data-driven clustering to assign a cluster label as surrogate for $C_b$ for each image (both composite and background-only images). While directly inferring a background cluster for an image containing an object that takes up the majority of the image may be difficult, in many cases access to wider crops containing mostly background may be possible, which is the case with sonar imagery.

For clustering the backgrounds, we propose using a two-stage process of either an autoencoder (AE) or an encoder trained with self-supervised contrastive learning (CL) followed by K-means on the encoder's output. The positive pair in the CL setting is defined as a randomly cropped background image (no less than 80%) and resized to match the original image.

Instead of K-means, we also implement an optimal transport K-means to encourage the cluster distribution to be uniform. That is for each K-means iteration, we run the Sinkhorn algorithm (Cuturi, 2013), using the Euclidean cost matrix between input data points and the centroids, with a uniform distribution for both, to yield a transportation plan that is cluster balanced. Due to the entropic regularization intrinsic to the Sinkhorn algorithm, each input point is distributed to multiple centroids. Before updating the centroids, new hard assignments are obtained from the highest mass in each row.

The encoder training and subsequent cluster assignment can be applied to all or a subset of the background-only images. Then, during the training of the masking network background-only images and composite images (or mostly background images from nearby crops) are assigned based on the nearest cluster to their encoding, yielding the label $C_b$. This produces the background-labeled images underlying the distributions required for Algorithm 1.

## 4 Data

While our goal is to perform real-world weakly supervised object segmentation on sonar images, datasets with ground truth labels are limited. We developed and initially validated our proposed method on artificial single-channel data constructed from combining real-world backgrounds—either textures (photographic images of materials with various textures (Brodatz, 1966) under controlled lighting) or seafloor images obtained from sonar (Cobb & Zare, 2014)—with objects from standard object classification datasets: standard MNIST, FashionMNIST (Xiao et al., 2017), and dSprites (Matthey et al., 2017). On these toy datasets we also ensure that the object is independently chosen from the background. Details and specific implementation of each of the toy datasets (Textures+MNIST; Textures+FashionMNIST; Textures+dSprites; and SAS+dSprites) are included in Appendix C.1.

We validate our approach on purely real datasets. The first is the AI4Shipwrecks dataset consisting of side-scan sonar (SSS) images (Sethuraman et al., 2024), which contain a variety of foreground objects corresponding to different sites that contain shipwrecks in Lake Michigan along with ground truth segmentation masks for the shipwrecks in the seabed. The second is the SAS-Clutter dataset consisting of small image snippets taken from high-frequency synthetic aperture sonar (SAS) collected by Naval Surface Warfare Center Panama City Division (NSWC PCD). The seafloor clutter objects in the images vary in size from

fractions of a meter to several meters in the longest dimension. The third is the Caltech-UCSD Birds-200-2011 (CUB-200-211) dataset (Wah et al., 2011), which consists of three-channel (RGB) color images of 200 species of birds on various backgrounds with segmentations.

## 4.1 AI4Shipwrecks

The AI4Shipwrecks dataset (Sethuraman et al., 2024) consists of 286 high-resolution SSS images with a variety of foreground objects corresponding to 28 distinct sites that contain shipwrecks in Lake Michigan along with ground truth segmentation masks for the shipwrecks in the seabed. The dataset is already divided 50/50 into training and test respecting the variety of sites. We further split the training data into 70% training and 30% validation.

For each sonar scene, we crop out the black strip in the middle (i.e., the blind spot directly below the sonar platform), and we flip the right side of the sonar image so that range increases consistently from right to left in images and the acoustic shadows all have the same orientation. Then, we crop 500 $128 \times 128$ images from each of the 143 images in the training set, creating 71,500 crops. As sonar data can have an extremely large dynamic range, we normalize each crop by dividing by its mean pixel value and then clipping any value larger than 16. We then resample each crop to $64 \times 64$. We use ground truth labels to label each crop as composite or background-only. Although this approach uses supervised information, the idea of weak supervision is that each crop could be quickly manually labeled, or one could use a statistical anomaly detection algorithm for the presence of objects in groups.

For testing, for each of the 143 test set images, we take $128 \times 128$ crops with 90%-overlap. Each crop is processed as above. The final segmentation mask is the mean across the overlapped masks. This is a convolutional approach, and the mean of the masks acts as a majority-vote segmentation, where the votes comes from the same pixel being in multiple overlapping crops.

## 4.2 SAS-Clutter

It is often necessary to identify and analyze small seafloor objects within SAS images. The SAS-Clutter dataset consists of approximately $10,000$ sonar images of underwater seafloor clutter objects collected using a high-frequency SAS from various coastal locations. The object image snippets were cropped from larger seafloor image survey data by first running a Reed-Xialoi (RX) based anomaly detector on the seafloor data and then manually selecting the snippets that actually contain objects (as opposed to other anomalies such as fish clouds, interference, or random noise). Seafloors are typically sparse, with very few objects, and infrequently change in composition/texture from one location to another. Thus, we make the assumption that crops taken from seafloor data adjacent to the selected objects will both be empty of objects and contain the same type of seafloor as its corresponding object. For each object image, we randomly selected a non-overlapping adjacent patch of seafloor to generate the background-only data. A validation set was created by manually drawing masks on a small random subset of approximately 200 objects. Furthermore, a set of approximately 2000 objects was held out for testing. Each object image was normalized by thresholding the pixel values at 16 times its mean pixel value and then rescaling to $[0, 1]$. Lastly, each image was resized to $64 \times 64$ using bilinear interpolation.

## 4.3 CUB-200-211

To illustrate the broader applicability of our approach across diverse imaging modalities, we also evaluate it on a dataset comprising natural images from the CUB-200-211 dataset (Wah et al., 2011), which consists of 11,788 images of birds from 200 species on various backgrounds. We chose this dataset to demonstrate that our work can also be applied to natural images, although the assumptions of independence of object and background are likely violated for birds with different habitats. Each image is min-max normalized such that all pixel values are between $[0, 1]$. We report the results for the separate testing set which consists of roughly half of the images; specifically, there are 5994 train images and 5794 test images. We further split the training data into 70% training and 30% validation. Additionally, when benchmarking against

other unsupervised object segmentation approaches, we follow Chen et al. (2019) and use their split, which consists of 10000 images for training, 1000 for testing, and the remaining 778 for validation.

## 5 Implementation Details

The AE or CL encoder (CLE) and K-means are trained on background-only images. The AE architecture is composed of 4 convolution layers each followed by a batch normalization layer and ReLU activation. The number of kernels in each layer are $64, 128, 256, 512$, respectively. A fully connected bottleneck layer of dimension $o = 20$ is used to represent the latent space. Similarly, the decoder is composed of one fully connected layer and 4 transpose convolutions with the same number of kernels as the encoder but in reversed order. We add a sigmoid layer at the end of the decoder. We use the Adam optimizer to optimize the AE, with a fixed learning rate of 0.0001, and batch size of 4096. The CLE follow the exact design of the encoder part in the AE but we increased the latent dimension to $o = 200$ for CUB dataset.

For the AI4Shipwrecks and SAS-Clutter datasets, a nearby patch of seafloor without any object can be passed to the AE/CLE + Sinkhorn K-means to infer the background of a foreground object image. The entropic regularization parameter in the Sinkhorn algorithm was set to 0.01. For CUB-200-2011, we train CLE and K-means and we use patches of size 25% of the image width and height taken from the composite images as background-only images: during CLE training, we use the ground truth masks to ensure that the patch does not contain any objects, but during training of the masking network, we use a random patch cropped from the input image. In this case, we use the ground truth to satisfy the assumption that representative background-only images are available; however we train the masking network without any ground truth knowledge. For the contrastive learning, we use augmentation techniques such as random cropping to assign positive pairs. We also use random horizontal flipping for the CUB dataset.

For AI4Shipwrecks and SAS-Clutter datasets, we choose clusters from the set $\{2, 3, 4, 5\}$ but we found that they perform very similarly so we only report the 4 clusters results. For CUB-200-2011, we train models with varying the number of clusters, choosing values from the set $\{5, 10, 15\}$. In both cases, the choice of cluster numbers is guided by the visual distinctiveness of example images in each cluster: we increase the number of clusters until further subdivision yields negligible improvement in visual separation.

We choose our segmentation model to be the U-Net architecture (Ronneberger et al., 2015b) with default hyperparameters. We choose D-adaptation optimizer to optimize U-Net (Defazio & Mishchenko, 2023). The learning rate for the U-Net is 1 and the batch size is 400. We selected the largest batch size the GPU used for training could handle. For the AI4Shipwrecks and CUB datasets, we run the training for 150 and 200 epochs, respectively. We choose the model with lowest validation loss $\mathcal{L}$ shown in equation 8. We implement our approach using the `PyTorch Lightning` (Falcon & The PyTorch Lightning team, 2019) framework. Only one GPU, a `Tesla V100-SXM2-32GB`, is used for training.

## 6 Results

We firstly present results for the real datasets: AI4Shipwrecks in Section 6.1, and SAS-Clutter in Section 6.2, and CUB-200-2011 in Section 6.3. Then we present the results of the synthetic objects on real sonar images SAS+dSprites in Section 6.4, and results on synthetic composite images Textures+MNIST/FashionMNIST/dSprites in Section 6.5. In the appendix, we show additional results on real images Appendix D, failure of benchmarks on single-channel images: Appendix C.4 for synthetic data and Appendix D.1.1 for the AI4Shipwrecks dataset.

Since our method is weakly supervised, it is noteworthy that our method achieves near-perfect performance (False Positive Rate $< 0.5\%$) for background images across all datasets and cluster numbers (except for the AI4Shipwrecks dataset, which has its own discussion). As a result, our primary focus will be on the segmentation of foreground objects. When ground truth masks are available we evaluate the segmentation performance on composite images using the following metrics: area-under-the-curve of the receiver operating characteristic (AUCROC), mean intersection of union (IoU) (also known as Jaccard index) with mask threshold at 0.5, and average precision.

## 6.1   AI4Shipwrecks

We first present visualizations of the input image, ground truth, estimated masking, and thresholded estimated masking for several images in Figure 2. We note that our masks often include more of the shadow, but the ground truth does not account for these shadows. Despite this, our model successfully recovers the ships, providing more precise segmentation in the second and third images. Further examples can be found in Figure 14 where the capturing of shadows is more pronounced. Additionally, we note that because the seabed sand moves, it may cover non-edge parts of the ships, making it reasonable for these regions to be segmented as background. This effect is even more pronounced in Figure 14, which demonstrates two key observations: first, our methodology captures the foreground shadow, an aspect not represented in the ground truth. Second, our model extracts 'foreground' objects that are not ships, such as rocks and coarse regions, as illustrated in Figure 15. Finally, in Figure 16 we show more examples of ships and small objects with relatively small shadows.

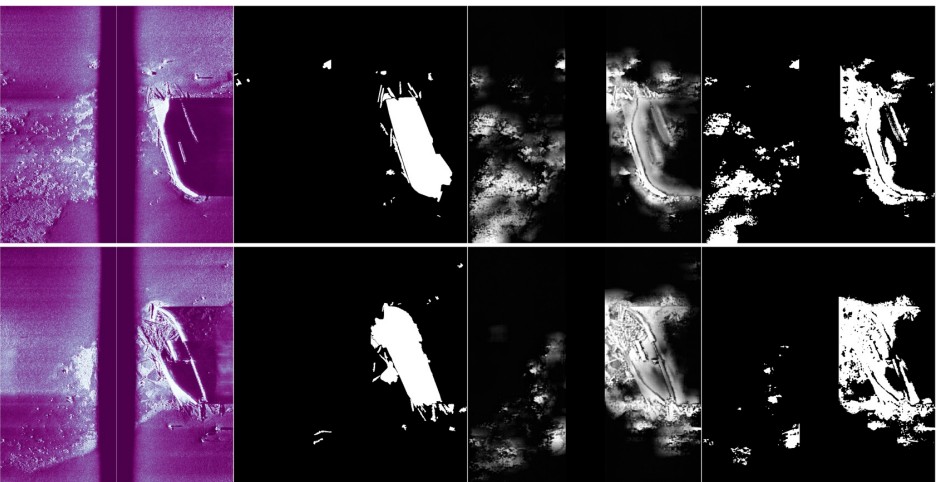

Figure 2: Segmentation model results on AI4Shipwrecks. Each row corresponds to a different image. From left to right: input images, ground truth masks, estimated masks, and thresholded estimated masks at 0.5. Images sites: Lucinda van Valkenburg.

### 6.1.1   Performance against Supervised Baselines

Given that our segmentation method includes additional foreground objects (e.g., rocks), captures shadows, and ignores portions of shipwrecks covered by sand—elements not considered in the ground truth—we believe a direct comparison with the provided ground truth may not fully reflect the model's performance. Nonetheless, we benchmark our method against supervised methods selected to reflect the variety of deep learning-based segmentation models used in the vision community: Yang (Yang et al., 2022a), ViT Adapter (Chen et al., 2022), DeepLabV3 (Chen et al., 2017), HRNet (Wang et al., 2020), U-Net (Ronneberger et al., 2015a), and Salient Object Detection, InSPyReNet (Kim et al., 2023). Following Sethuraman et al. (2024), we aggregate the images per site and report the IoU for each cite in Table 2. Our unsupervised method has comparable or superior average performance to 4 supervised methods, but is outperformed by supervised U-Net and SOD. We also note that across the sites our performance co-varies with the best method. We outperform other methods on sites where the supervised methods also fail. For example, on the 'James Davidson' site, one portion shown in Figure 17, false alarms are evident, where parts of the background are incorrectly identified as foreground, with no clear rationale behind these misclassifications. We believe this issue arises because the models have not encountered such a background during training and, as a result, mistakenly segment it.

| Site | Yang | ViT-Adapter | DeepLabv3 | HRNet | U-Net | SOD | Ours | Our rank |
|------|------|-------------|-----------|-------|-------|-----|------|----------|
| Artificial Reef | 0.002 | 0.003 | 0.003 | 0.011 | 0.006 | 0.017 | **0.05** | 1 |
| Barge | 0.471 | 0.412 | 0.670 | 0.703 | 0.736 | **0.775** | 0.52 | 5 |
| Corsair | 0.231 | 0.542 | 0.531 | 0.293 | 0.564 | **0.583** | 0.43 | 5 |
| Corsican | 0.056 | 0.088 | 0.042 | **0.174** | 0.116 | 0.077 | 0.162 | 2 |
| James Davidson | 0.000 | 0.000 | 0.031 | 0.067 | 0.074 | 0.091 | **0.092** | 1 |
| WH Gilbert | 0.276 | 0.006 | 0.658 | 0.641 | 0.726 | **0.749** | 0.623 | 5 |
| Haltiner Barge | 0.031 | 0.368 | **0.459** | 0.448 | 0.242 | 0.442 | 0.357 | 5 |
| Lucinda van Valkenburg | 0.367 | 0.127 | 0.645 | 0.641 | 0.641 | **0.713** | 0.542 | 5 |
| Monohansett | 0.020 | **0.496** | 0.034 | 0.207 | 0.042 | 0.467 | 0.361 | 3 |
| Monrovia | 0.478 | 0.469 | 0.411 | 0.566 | 0.572 | **0.572** | 0.487 | 4 |
| Shamrock | 0.102 | 0.354 | 0.151 | 0.003 | **0.428** | 0.001 | 0.143 | 4 |
| WP Thew | 0.161 | 0.176 | 0.419 | 0.437 | 0.545 | **0.580** | 0.391 | 5 |
| Viator | 0.563 | 0.631 | 0.667 | 0.646 | 0.646 | **0.776** | 0.521 | 7 |
| Average | 0.21±0.20 | 0.28±0.22 | 0.36±0.27 | 0.37±0.26 | 0.41±0.28 | 0.45±0.30 | 0.36±0.19 | 4.00±1.78 |

Table 2: Per-site IoU comparison between our method and supervised benchmark baselines on the AI4Shipwrecks dataset. Bold indicates the best performance per site.

### 6.1.2 Unsupervised Segmentation Benchmarks

We attempted to apply unsupervised object segmentation methods as benchmarks. We choose ILSGAN (Zou et al., 2023) as it is the best performing method among the generative-based methods, ReDO (Chen et al., 2019) as it makes the minimal assumption about the foreground objects (e.g., it does not assume a minimal object size), and finally the work by Savarese et al. (2021) for its unique attributes (i.e., no pretrained models, no adversarial training, and no need for data labels). For all, we use the same pre-processing steps explained in Section 4.1 with the same patch size.

For ILSGAN and ReDO, we adapt the model architectures to accept one channel instead of three, and use the same hyperparameters as in the original works. We found that the networks produce either trivial masks (i.e., all ones or all zeros) or masks without any meaningful segmentation. The work ReDo by Chen et al. (2019) converged to either almost all-zero masks or all-ones masks as shown in Figure 18 in the appendix. The foreground mask typically contains either diffuse low-intensity values or small, high-intensity regions. These mask patterns appear to be leveraged by the model to promote statistical dependency—i.e., mutual information—between the input noise vector and the resulting generated image. We use their default setting of generating two masks. Additionally, to ensure a fair comparison, we also tried adding a supervised loss for background-only images, the mean squared loss for the generated mask, which is minimized if the generated mask is all zeros, as in equation 7. Figure 19 shows examples for background and composite patches where our MSE loss drives the foreground mask to be completely all-zeros. For ILSGAN (Zou et al., 2023), the trained model was able to generate realistic images; however, the corresponding masks almost all zero. In other words, the model degenerates and ignores the possible presence of the shipwreck objects shown in Figure 20 in the appendix. This is not surprising as the 'foreground' is very similar to the background.

Finally, we attempted to use the method by Savarese et al. (2021); since it is a model-free method, we applied two schemes: the first inputs the whole sonar image and the second uses a patch-based scheme with 50% overlap. However, both schemes provide nonsensical masks as expected for an unsupervised method designed to work on color images with the object at the center. Results are shown in Figure 21 in the appendix. Although we show only two examples, this observation is consistent across the testing images.

Admittedly, the hyperparameters selected in prior work, which work when trained on colored natural images, may not be optimal; however, for unsupervised segmentation it is not clear how to conduct the required hyperparameter search. Even with the weak supervision, we hypothesize that these methods may simply be poorly suited for single channel images, such as sonar images, that lack the color and optical cues of foreground and background found in natural imagery. To test this hypothesis, in addition to sonar images, we also benchmark other unsupervised segmentation methods using our synthetic datasets (see Appendix C.4).

## 6.2 SAS-Clutter

Figure 3 visualizes a random sampling of the test set seafloor clutter objects, their learned masks, and the resulting pixel-wise multiplication of the mask and the object images. In each case, the model successfully masks the bright object highlight. Furthermore, in cases where a shadow exists, it can be seen that the masks extend to encompass them as well. However, while the highlight masks tend to be dilated in size, encompassing the entire object, the shadow masks tend to be slightly eroded. As the vast majority of the object images have no ground truth segmentation mask, comparison with supervised methods is impossible. As with the AI4Shipwrecks dataset, we attempted to compare performance with the three unsupervised approaches ILSGAN (Zou et al., 2023), ReDO (Chen et al., 2019), and the in-painting method by Savarese et al. (2021). Both GAN approaches (ILSGAN and ReDO) failed to produce usable masks. The in-painting approach by Savarese et al. *was* able to produce reasonable masks; however, at convergence, they covered only the brightest highlights of each image. We computed the IoU over approximately 200 hand-drawn ground truth masks for our approach and the in-painting approach by Savarese et al.. For the in-painting approach we report both the best IoU over all iterations and the IoU at convergence. Our approach produced an IoU $0.46 \pm 0.18$ while the in-painting approach gave $0.36 \pm 0.14$ and $0.21 \pm 0.15$.

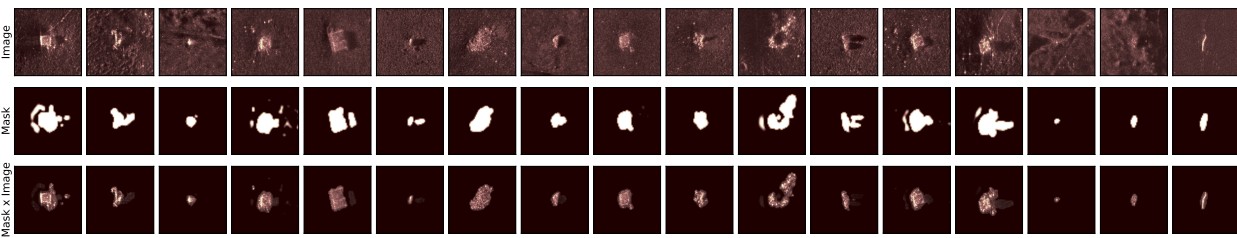

Figure 3: Segmentation model results on SAS-Clutter object images. Each column consists of a object snippet image, its estimated segmentation mask, and the masked object.

## 6.3 CUB-200-2011

To qualitatively assess segmentation performance on CUB, we show 16 images from the testing set along with their segmentations and the birds masks in Figure 4. In Figure 5, we show a variety of counterfactual images.

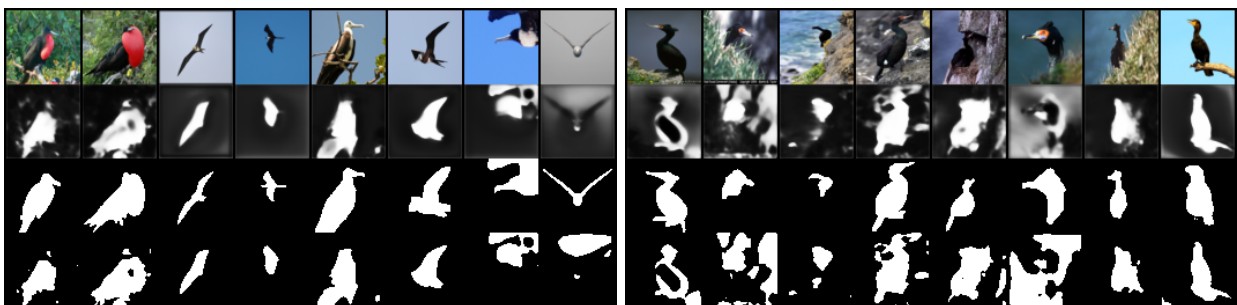

Figure 4: Results of the segmentation model for CUB. From top to bottom: input images, estimated masks, true masks, and thresholded estimated masks at 0.5.

We notice that the majority of the generated counterfactual images do look like real images. However, there are interesting cases where the masking network, in addition to birds, captures other objects. For example, it captures tree branches (e.g., third row, 6th and 7th columns), rocks (e.g., third row, 14th column), wires (first row, last column), cars (first row, 8th column). These objects can be still considered as foreground. Also, we can see some unexplained failure such as the 10th and 12th pictures in the first row, where for the former the counterfactual does not appear to differ from the input image and for the latter, the counterfactual does not

look like anything in the CUB dataset. To delve into more examples, we show the generated counterfactual

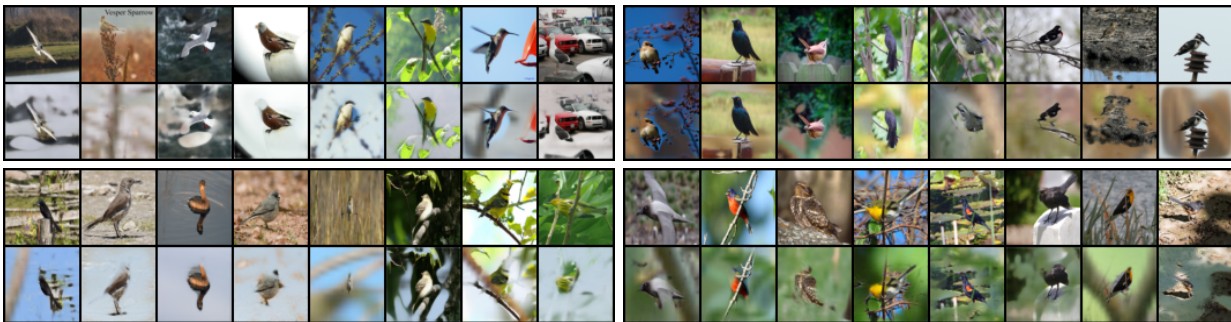

Figure 5: Generated counterfactual images based on test images using 10 clusters. Each block is composed of two rows and 8 columns. Each block shows the original input images in the first row and the corresponding counterfactual images in the second row. The counterfactual are created by alpha-blending the real input images with randomly selected backgrounds. Each second row in the blocks should ideally be composed of the same background clusters. An example of the obtained clusters is shown in Figure 22 in the appendix.

images using the corresponding source composite images (first column) and the source background images (first row) in Figure 6. For quantitative results, we show in Table 3 the IoU metric for different cluster numbers under AE+K-means and CL+K-means settings (other metrics are in Table 7).

Table 3: Masking Network Performance (IoU) for Different Background Clusterings.

| Number of Clusters | AE+K-means | CL+K-means |
|:---:|:---:|:---:|
| 5 | 20% | **30**% |
| 10 | 24% | **30**% |
| 15 | 24% | **26**% |

### 6.3.1 Benchmark Comparison

We also include a comparison between our method and other related work that does not require pretrained networks. Here, we use 5 clusters. The model is different than the one previously discussed, as the benchmark (Chen et al., 2019) uses more training data, i.e., 10,000 instances compared to the approximately 5,000 instances used in Table 3. As shown in Table 4, with the additional training data, our IoU performance increases by 6 percentage points; however, our approach is outperformed by most of the models.

Table 4: IoU scores (%) on the CUB dataset across different methods. Bold indicates the best score. From left to right, each work corresponds to Singh et al. (2019), Chen et al. (2019),Bielski & Favaro (2019),Hung et al. (2019),Savarese et al. (2021),Yang et al. (2022b), He et al. (2022), Zou et al. (2023), respectively.

| FineGAN | ReDO | PerturbGAN | SCOPS | Savarse et al. | Yang et al. | GANSeg | ILSGAN | Ours |
|:---:|:---:|:---:|:---:|:---:|:---:|:---:|:---:|:---:|
| 44.5 | 42.6 | 38.0 | 32.9 | 55.1 | 69.7 | 62.9 | **72.1** | 35.6 |

Our model's modest performance is characteristic of its objective: generating plausible counterfactuals. The masking network finds a solution not aligned to the ground truth due to three main reasons: it can create plausible counterfactuals by taking portions of images that create bird-like silhouettes, as seen in Figure 6 rows 4, 6, 9, 10, and 11; it can ignore portions of birds bodies that can vary in color, such that when blended with background they are still reasonable representations (perhaps of different species); and thirdly, the strong contextual dependency between foreground objects and their backgrounds. When a strong dependency exists, a cleanly segmented foreground object often creates a statistically improbable image when placed in a new context, resulting in a high loss. Instead of this, our model strategically finds masks that include 'contextual anchors' from the source background to preserve local realism and create a more plausible

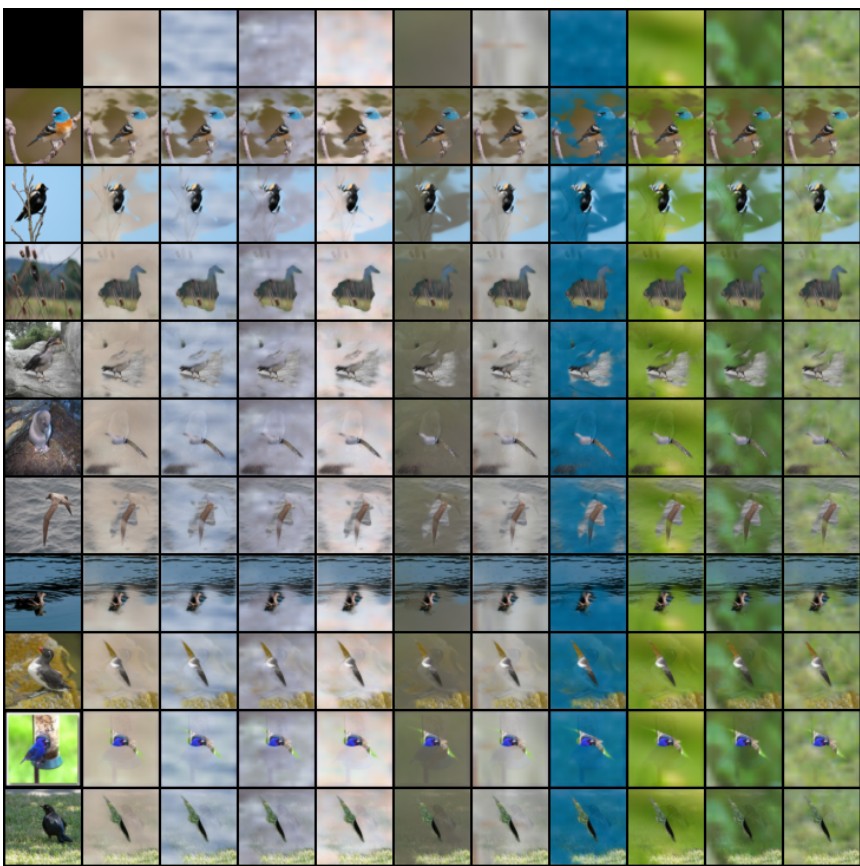

Figure 6: Generated counterfactual images based on test images using 10 clusters. The first row shows the source background image where each image comes from a different background cluster, and the first column is the source composite image. Each cell in the grid (except the first row and column) represents the alpha-blending of the corresponding source background and source composite images. Each column in this figure shows a background that corresponds to the source background at the top, and every row shows a foreground object that corresponds to the source foreground object at the most left, which demonstrates, generally speaking, that our method is preserving both background and foreground information. Also, the general appearance of the counterfactual images do correspond to the original data. It is interesting to see that in the last three rows, the model is able to craft counterfactuals with bird-like images that are not strongly related to the input composite images.

final image. For instance, consider a task where the source image is a bird native to a dense forest, and the target distribution is characterized by backgrounds that are mostly open sky with a few branches. To minimize the statistical distance, the model will not just segment the bird; it will segment the 'bird-on-branch' unit together. A forest bird floating against a plain sky is a statistical anomaly, but by including the branch, the model preserves a high-probability visual pattern. Similarly, for the same target distribution, creating a silhouette of a flying bird out of any composite image will create a plausible counterfactual with the open sky background. Together, these masks create a counterfactual distribution that better align with the target distribution, thereby achieving a lower loss, but do not align with the ground truth mask of the bird alone. While this masking behavior leads to imperfect segmentation on CUB, it highlights our method's key strength—it is ideally suited for domains where the foreground and background are decoupled. As shown for the sonar datasets, man-made objects like shipwrecks (artificial reefs, abandoned crab/lobster traps, or unexploded ordinances or mines) do not correlate with the surrounding seabed. In this setup, where contextual priors are not a factor, our method excels, proving competitive even with supervised approaches.

### 6.4 SAS+dSprites

We present both quantitative and qualitative results for sonar dataset that combines real-world seafloor SAS imagery from Cobb & Zare (2014) with synthetic objects from the dSprites dataset. The seafloor contains six texture categories: sand ripple, rock, coral, sand, mud, and seagrass. Figure 7 illustrates the segmentation performance, demonstrating that the proposed method generates high-quality segmentation masks. For this dataset, the method does not produce 'sharp' masks. Table 5 provides the performance metric values for different numbers of background clusters. The results indicate that the best performance is achieved when using four clusters.

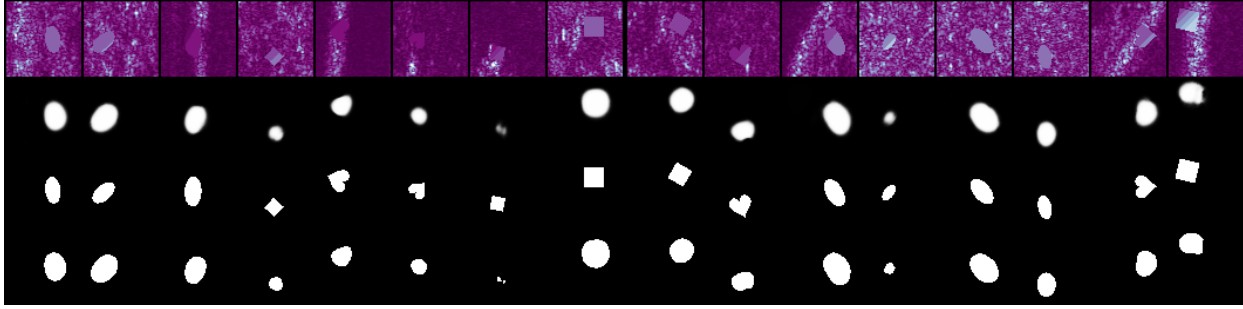

Figure 7: Results of the segmentation model for synthetic objects on real sonar images. From top to bottom: input images, estimated masks, true masks, and thresholded estimated masks at 0.5.

Table 5: Performance across Different Numbers of Clusters for Synthetic Objects on Real Sonar Images

| Clusters | Avg. Precision | IoU | AUC-ROC |
|---|---|---|---|
| 3 | 60% | 30% | 90% |
| 4 | 82% | 57% | 97% |
| 5 | 22% | 20% | 87% |

### 6.5 Textures+MNIST/FashionMNIST/dSprites

We provide both quantitative and qualitative results for three datasets: MNIST, Fashion MNIST, and dSprites. Initially, we trained the segmentation model using true clusters, defined by the texture label. Subsequently, we trained the segmentation model based on AE clustering. We begin by presenting quantitative results, followed by showcasing qualitative outcomes for the best-performing models on each dataset. We show eight testing images, the estimated masks, the true masks, and the thresholded estimated mask based on 0.5 in Figure 8 using 10 clusters obtained from the AE+K-Means.

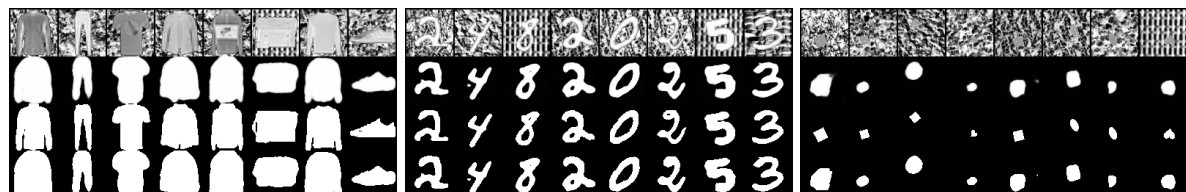

Figure 8: Results of the segmentation model for Fashion MNIST (left), MNIST (middle), and dSprites (right). Top to bottom: input images, estimated masks, true masks, and estimated masks thresholded at 0.5

Figure 9 shows the quantitative performance results for the three datasets with respect to the number of clusters from K-means along with performance using the true texture classes indicated by stars. Qualitative

results for the masks using the true texture classes in provided in Figure 11 in the appendix. The performance

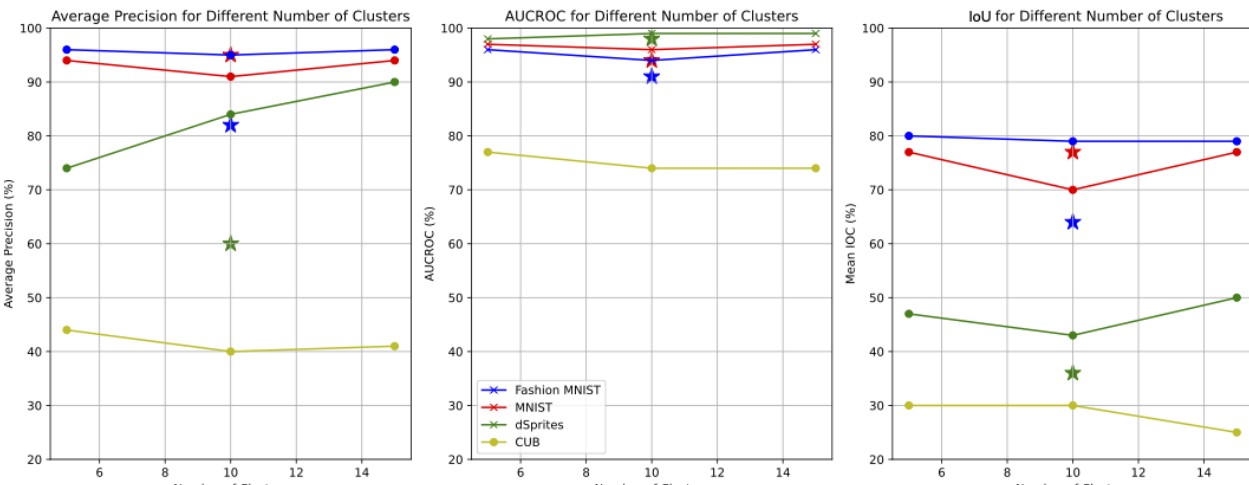

Figure 9: The percentage of Average Precision, AUCROC, IoU values (y-axis) for the four datasets, as indicated by the legend. The x-axis represents the chosen number of clusters for the K-means model based on AE/CLE features. The star symbols above the x-axis values, specifically at the position of 10, emphasize the Average Precision and AUCROC corresponding to the ground truth clusters.

of the model on Fashion MNIST, which has relatively large objects with polygonal-like shapes, is high with all clusters performing relatively the same. The model performs similarly on MNIST when utilizing any of the estimated clusters, except for a slight drop when using the 10 clusters. We notice, however, that the performance of the true clusters is worse when compared with the estimated ones, which is evident in the qualitative results as well (see Figure 11 in the appendix). The model's performance on the dSprites dataset is comparatively lower, which we primarily attribute to the small size of objects and intensity variation. Despite this, the qualitative results depicted in Figure 8 show that as in the case of sonar data, the masks are not 'sharp'. It is noteworthy that the mask size is proportional to the object size. Notably, when employing AE-based clustering with a cluster number equal to 10, the dataset achieves the highest average precision at 83%, compared to 60% when using true clusters.

To further discuss the reason for the superior performance for estimated clusters versus the true texture classes, we show the clustered background using AE+K-means along with the corresponding true clusters in Figure 10 in the appendix. We notice that the K-means clustering creates clusters out of the same true cluster, and each cluster has a different phase. For example, in the case of 5 clusters, the brick cluster is divided into two out-of-phase clusters. It also groups together distinct true backgrounds into one cluster. Another example, in the case of 5 clusters, the first row, which represents a cluster, shows a mix of two true clusters. Consequently, images with essentially the same content but differing phase are not assigned to the same cluster.

## 7 Discussion

We introduced a framework for weakly-supervised foreground object segmentation within a dataset comprising both foreground and background images. Our approach leverages background images to formulate a straightforward loss function that avoids degenerate solutions and offers a more straightforward training scheme compared to generative and ViT-based models. Our approach can be applied without relying on large models pretrained on natural imagery. Methods based on such models often cannot be directly applied to other data modalities. In contrast, our approach can be directly applied to other modalities such as sonar data, where the image encodes intensity of the echo. The distinctive combination of our dataset and framework sets our work apart.

We demonstrate the limitations of related work by applying two works by Bielski & Favaro (2019) and Xie et al. (2022) to multiple synthetic datasets. Both methods failed to converge to a solution where the segmentation results are acceptable. This is due to the lack of a well-pretrained model (for the work done by Xie et al. (2022)) and having diverse classes of foreground objects (for the work done by Bielski & Favaro (2019)).

In addition to sonar data, we also demonstrate relatively good performance on natural images using the CUB dataset, but note that counterfactual generation in cases with strong object-background dependence limits performance. During training of the masking network, random patches can be taken from the input image to assign a background cluster for the conditional divergence loss. This demonstrates that our approach can also be used for datasets composed only of composite images (as long as the images are not tightly cropped around the objects).

We have assumed only access to composite images and background-only images. If we also have access to the ground truth masks, one can incorporate 'cut, paste and learn' approaches that use known ground truth masks to help train segmentation models (Dvornik et al., 2018; Dwibedi et al., 2017; Fang et al., 2019) as additional synthetic supervision to be combined with weakly supervised real cases. However, we note that the location selection problem involved in 'cut, paste and learn' may not be as crucial in image domains such as sonar. Nonetheless, the various blending techniques by Dvornik et al. (2018) could alleviate some of the artifacts in counterfactuals.

Although one of the main contributions of our work is the avoidance of pretrained models, they can be used to extract reliable initial heuristic cues of where the foreground object is. This can be useful in extracting background patches that are used to train the clustering AE or to help guide the masking network.

The main limitation of our approach is the reliance on an appropriate background clustering. We observed that each dataset achieves optimal performance with a different number of clusters, constructed under varying settings. Selecting the correct number of clusters is a challenging task, but could be guided by visual inspection of images assigned to potential clusters. Notably, the optimal clustering does not have to correspond to true background classes. On the dSprites synthetic dataset, using the clusters actually outperformed the background classes corresponding to the texture class.

## 8 Conclusion

We have investigated a novel technique for weakly supervised segmentation based on the assumptions that an object's appearance is independent of the background imagery and there exists sufficient variety across clusters of backgrounds. Given these assumptions, we can avoid a trivial solution when seeking a masking that creates counterfactuals that superimpose a masked object obtained on a different background by minimizing a divergence conditioned on the background cluster. The method, which does not use any pretrained network, performs well on real-world sonar images of shipwrecks and synthetic images where the assumptions hold and other unsupervised and weakly supervised segmentation methods fail. It achieves modest performance on natural images where only the second assumption holds. Future work can investigate possibilities for semi-supervised approaches that incorporate limited ground-truth masks.

**Acknowledgments**

Research at the University of Delaware was sponsored by the Department of the Navy, Office of Naval Research under ONR award numbers N00014-21-1-2300 and N00014-24-1-2259. This material is based upon work supported by the National Science Foundation under Award No. 2108841. This research was also supported in part through the use of DARWIN computing system: DARWIN – A Resource for Computational and Data-intensive Research at the University of Delaware and in the Delaware Region, which is supported by NSF under Grant Number: 1919839, Rudolf Eigenmann, Benjamin E. Bagozzi, Arthi Jayaraman, William Totten, and Cathy H. Wu, University of Delaware, 2021, URL.

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

# A  Literature Review

State-of-the-art approaches in unsupervised object segmentation area can be divided into three main categories—the first is generative-based (e.g., GANs), the second is based on vision transformers (ViT), and the third is class activation map (CAM). Table 6 summarizes approaches based on their reliance on pretrained models, GAN training—which can create unstable learning dynamics— object class labels, and extensive hyperparameter tuning to balance composite losses.

## A.1  Generative-based Models

The work by Remez et al. (2018) is similar to ours in that it proposes a weakly-supervised approach that relies on a discriminator to assess the statistical similarity between synthetic images with counterfactual backgrounds and real images. Notably, their discriminator relies on a pretrained object classifier. Furthermore, the masking network input is based on the features from a pretrained object recognition network that outputs a bounding box for the foreground object. The necessity of using pretrained models and a network to generate a bounding box are the main limitations of this work.

Ostyakov et al. (2018) proposed a generative adversarial network (GAN) that takes a composite image as input and inpaints the foreground region with a background consistent with the original background. Subsequently, the segmented object is pasted onto a different background to generate a counterfactual image. Their model involves a segmentation network that aims to produce inpainted and counterfactual images realistic enough to deceive the discriminator. In addition to two discriminator networks, their architecture includes three auxiliary networks: a segmentation network, an inpainting network, and an enhancement network—the latter designed to correct artifacts resulting from inaccurate object masking. To avoid trivial segmentation masks, the first discriminator distinguishes between inpainted composites and real background images, while the second discriminator differentiates counterfactual images from genuine composite images. Their total loss is extremely complicated; it consists of 11 different losses.

Bielski & Favaro (2019) proposed a GAN-based, unsupervised approach to segment foreground pixels. The GAN produces mask, foreground, and background images. The mask is used to alpha-blend the background and foreground back together. Trivial solutions for this work result when (i) the generated background and foreground are identical, so any arbitrary mask would still give a real image, (ii) Either the foreground or the

Table 6: Literature Review on Weakly Supervised Foreground Object Segmentation: All of the reviewed work, depends on at least one undesired attributes (pretrained model, GAN training, data labels, Extensive Hyperparameters) that hinders applying them to data-scarce domains such as sonar images.

| Category | Reference | Pretrained Models | GAN Training | Data Labels | Extensive Hyperparameters |
|---|---|---|---|---|---|
| **Generative-based** | Remez et al. (2018) | ✓ | ✓ | | ✓ |
| | Ostyakov et al. (2018) | | ✓ | | ✓ |
| | Bielski & Favaro (2019) | | ✓ | | ✓ |
| | Chen et al. (2019) | | ✓ | | |
| | Singh et al. (2019) | | ✓ | | ✓ |
| | He et al. (2022) | | ✓ | | ✓ |
| | Yang et al. (2022b) | | ✓ | | ✓ |
| | Zou et al. (2023) | | ✓ | | ✓ |
| | Sauer & Geiger (2021) | ✓ | ✓ | | ✓ |
| | Li et al. (2023) | ✓ | | | |
| **Vision-Transformers** | Dosovitskiy et al. (2021) | ✓ | | | |
| | Hamilton et al. (2022) | ✓ | | | |
| | Wang et al. (2023b) | ✓ | | | |
| **CAM-based** | Zhang et al. (2018) | ✓ | | ✓ | |
| | Li et al. (2018) | ✓ | | ✓ | ✓ |
| | Hou et al. (2018) | ✓ | | ✓ | |
| | Jiang et al. (2019) | ✓ | | ✓ | |
| | Chang et al. (2020) | ✓ | | ✓ | |
| | Lee et al. (2021) | ✓ | | ✓ | |
| | Wu et al. (2021) | ✓ | | ✓ | |
| | Xu et al. (2022) | ✓ | | ✓ | |
| | Xie et al. (2022) | ✓ | | ✓ | |
| **Others** | Hung et al. (2019) | ✓ | | | ✓ |
| | Savarese et al. (2021) | | | | ✓ |

background is ignored (i.e., all-foreground or all-background), which results in an either all-zero or all-one mask. To prevent the latter, the hinge loss on the average mask value is deployed. To prevent the former, a small random spatial shift of the composite image and mask are applied before alpha-blending to create a counterfactual shift in the resulting synthetic image. This shift penalizes having poor generated masks with identical foreground and background by creating a statistical divergence between shifted counterfactuals (i.e., the generated image) and real images, which the discriminator can detect. A drawback to this approach is the number of different loss functions (size, binary, gradient penalty, discriminator output norm) and hyperparameters required to prevent degenerate solutions.

Building on the layered-GAN approach of the previous work (Bielski & Favaro, 2019), the method by Yang et al. (2022b), used more general perturbations of mask and foreground via affine transforms and added mutual information objectives between the random codes for foreground generation and the resulting mask and synthetic image. There are two random codes for the two GANs (GAN-1 and GAN-2). The first code is shared and used by GAN-1 for generating backgrounds and GAN-2 for generating foregrounds and masks in conjunction with the second code which is used only by GAN-2. Maximizing the mutual information between the rendered synthetic image and the second code prevents producing all-zero masks—if the mask is all-zeros then the image will be equivalent to the generated background image which has zero mutual information with the second code. On the other hand, maximizing the mutual information between the mask and the second code encourages mixed-value masks (i.e., variety of zeros and ones). The reason behind this is that since the map is deterministic, the mutual information is equivalent to the entropy of the mask. Finally, a masking network is jointly trained based on the data generated by the GAN model. This approach could be adapted to our setting by avoiding the GANs and considering the mutual information between the mask and synthetic image and the original composite image as an additional objective.

A similar work by Zou et al. (2023), ILSGAN, introduced an information-theoretic regularization which maximizes the independence (minimizes an upper bound on mutual information (Cheng et al., 2020)) between (1) the visible foreground (i.e., foreground pixels where the mask values are ones) and invisible background

(i.e., background pixels where the mask values are zeros), (2) invisible foreground (i.e., foreground pixels where the mask values are zeros) and visible background (i.e., background pixels where the mask values are zeros). Additional losses are added to encourage non-zero and binary masks. While this work addresses the issue of preventing degenerate solutions, a notable drawback is the utilization of multiple loss functions which requires a careful fine-tuning for the weight of each loss.

Chen et al. (2019) proposed ReDO as an unsupervised object segmentation method based on the assumption that foreground texture and background content are statistically independent (e.g., a flower can change color regardless of its background). Their model consists of a segmentation network that predicts a soft mask and a generator network that produces a new foreground texture conditioned on a Gaussian noise vector. The final output image is formed by alpha-blending the generated texture and the original image using the predicted mask. To avoid degenerate masks (e.g., all-zero or full-coverage), they introduce a mutual information objective: a separate network is trained to recover the original noise vector from the generated image, encouraging the mask to preserve informative regions relevant to texture generation.

Singh et al. (2019) proposed FineGAN, a hierarchical gan that (i) generates the background image by using a vanilla GAN that only focuses on the background patches obtained using off-the-shelf object detection models, (ii) generates the shape and texture hierarchically by imposing a parent-child relationship between the generated shape and texture, where a shape (i.e., the parent) can have many textures (i.e., the child). This is realized by conditioning the generated texture on the code used to generate the mask. Note that for generating the shape, infoGAN loss where it maximizes the mutual information between the code and the generated image as there is no access to real images that has no textures.

The work by Sauer & Geiger (2021) introduced a Counterfactual Generative Network (CGN), leveraging the concept of learning Independent Mechanisms (IMs) to establish a causal structure for mask, foreground, and background. Each was modeled by fine-tuning a distinct pretrained BigGAN. To ensure the resulting composite image is real, they minimize the MSE loss between the output of pretrained frozen BigGAN and the composition process. To ensure the generated mask is not trivial, hinge and entropy losses are deployed. For the background loss, the predicted saliency is minimized which leads to the foreground object's disappearance. For background generation a pretrained U2Net is used to detect the saliency. Overall, this approach heavily relies on pretrained models and necessitates the incorporation of specific losses to prevent degenerate or trivial solutions.

He et al. (2022) proposed GANSeg, a generative model that synthesizes foregrounds, backgrounds, and segmentation masks in a hierarchical manner. Foreground and mask generation are conditioned on spatial embeddings derived from Gaussian noise, which encode part-specific locations. Each object is composed of multiple segments with each segment having its own mask; the final object mask is obtained by aggregating these individual masks. To avoid degenerate solutions where the masks collapse to all-zeros or all-ones, a hinge loss is applied to constrain the overall mask area. Additionally, a geometric consistency loss encourages each segment's mask to be spatially concentrated around its designated location, promoting more structured and interpretable mask generation.

The work by Li et al. (2023) proposes a training-free image segmentation method that leverages pretrained generative inpainting models, such as latent diffusion models (Rombach et al., 2022). Given an initial coarse mask (e.g., a bounding box), the image is first inpainted to hallucinate the background in place of the masked foreground. A contrast map is then computed by comparing the original and inpainted images, using both pixel-wise color differences and L-2 distances in the feature space of a pretrained encoder that preserves spatial resolution. K-means clustering (with $k = 2$) is applied to this contrast map, and the cluster with the higher mean contrast is interpreted as the refined foreground. This enhanced mask is then used in an outpainting step, in which the background is masked and reconstructed using the generative model. The contrast between the original image and this outpainted version is again analyzed to further refine the segmentation mask, following the same contrastive and clustering strategy. This process is iterated multiple times to resolve false positives and false negatives in the initial segmentation. While this method is notable for requiring no supervised training, its reliance on powerful pretrained generative models may limit its applicability in domains where such models are unavailable or underperforming. Moreover, its segmentation quality is contingent on the inpainting fidelity and the robustness of contrast-based clustering

To summarize, the presented works rely on either pretrained models or complex combinations of loss functions—involving training GANs, dependencies that our approach avoids.

## A.2 Vision-Transforms-based Models

The vision transformer (ViT) (Dosovitskiy et al., 2021) treats patches of input as tokens whose representations are processed through layers involving multi-head attention, and can be adapted to unsupervised segmentation. For example, the work by Wang et al. (2023b) used the ViT model DINO (Caron et al., 2021) to obtain features from non-overlapping patches of the input image. These features were used to construct a graph that represents the similarity between the patch-wise features. The normalized cut (N-cut) (Shi & Malik, 2000) algorithm was then applied, which was followed by CRF (Lafferty et al., 2001) for edge enhancement. A similar work was proposed by Wang et al. (2023a) which uses multiple N-cuts on masked images to detect multiple objects. Again, post-processing, such as conditional random field (CRF) (Lafferty et al., 2001) or bilateral solver (Barron & Poole, 2016), is required to get pixel-wise masks from the coarse patch-wise segmentation.

ViTs require large amounts of training data (Dosovitskiy et al., 2021), which can hinder the performance of the presented works if sufficient data is unavailable or if the data domain differs significantly from that of the pretrained ViT models. Even with sufficient data for ViT training, the demand for significant computational resources remains, posing an additional constraint (Irandoust et al., 2022). All the works mentioned below depend on pretrained models, which is limiting for specialized image modalities.

The work by Hamilton et al. (2022) proposed an unsupervised semantic segmentation approach that distills knowledge from pretrained feature extractors. A frozen backbone (e.g., DINO-ViT) is used to extract dense pixel-wise features from input images. A trainable segmentation network maps these features to lower-dimensional embeddings representing per-pixel segment assignments. During training, a mini-batch is constructed for each sample by including its K-nearest neighbors in the feature space. The model then samples a neighbor image and computes a cosine similarity matrix between the pixel features of the target and the neighbor. The segmentation output is encouraged to be similar for pixels with high feature similarity and dissimilar for pixels with low similarity.

## A.3 Class-Activation-Map-based Models

Much work on weakly-supervised segmentation is based on class-activation maps (CAM) derived from pretrained convolutional object classification networks (Kumar Singh & Jae Lee, 2017; Wei et al., 2017; Zhang et al., 2018; Li et al., 2018; Hou et al., 2018; Jiang et al., 2019). We stress the fact that our model does not require labeled data (for instance, class information for composite/background images). Performance was initially limited as segmentation based on activation maps often focused on a few parts of the objects (Kumar Singh & Jae Lee, 2017; Wei et al., 2017). To mitigate this problem, Zhang et al. (2018) used a backbone fully-convolutional network to extract initial CAMs and then used two parallel-branches, each a CNN classification network, which took the initial CAM as an input. Parts of the input CAM of the first classifier are erased with the guidance of the object localization map from the second classifier. This encourages the first classifier to seek other parts of the image to do the classification. Thus, this results in learning a more coherent/complete feature map that can be used for object segmentation. Similarly, Li et al. (2018) proposed two networks, where one network uses the parts that are neglected by the first one for the classification task which results in expanding the activation map to other related parts. The work by Hou et al. (2018) introduced an approach for adversarial learning during supervised classification training which encourages the CAM to expand from covering only part of the target objects while penalizing it for covering background regions spuriously using the background prior (i.e., regions with low activations defined as background, regions with high activations defined as foreground and regions in-between defined as potential foreground) obtained from the initial map. Following a similar aim, Jiang et al. (2019) observed that the attention maps produced by a classification network changes focus on different object parts throughout the learning. They saved copies of the model each epoch and then used them to produce CAMs. At the end, they fused all the resulting CAMs using a pixel-wise max operation, which enhanced the final CAM. All previously mentioned work in this paragraph depends on pretrained models.

The work by Chang et al. (2020) joined supervised learning with self-supervised learning to enhance CAM. First, a CNN is trained to classify images according to their true labels. After that, the internal representation of the input image is used as a feature representation. The aggregated feature representations are then used to cluster each class into pseudo-labels (e.g., person label can be divided into child and adult). Then, another classifier is trained to classify the pseudo-labels/clusters of the feature representations while backpropagating the loss to fine-tune the original CNN to improve the feature representation. After that, the thresholded resulting CAM is used for semantic segmentation.

The work by Lee et al. (2021) uses information obtained from an off-the-shelf, pretrained saliency detector (trained in supervised or unsupervised fashion) to improve the CAM semantic segmentation. They minimize the loss between the estimated saliency map based on CAMs and the one obtained from off-the-shelf models while minimizing the classification loss.

Wu et al. (2021) initially extracted the class-specific CAMs obtained from an off-the-shelf model (i.e., for each input image, $K$ CAMs are generated where $K$ are the number of classes). Then, a self-attention mechanism is applied to all $B \times K$ CAMs , where $B$ is the batch size, from the batch, enabling the model to capture both spatial dependencies within each CAM (intra-CAM) and semantic relationships across different CAMs (inter-CAM), thereby improving attention consistency and completeness. The output is then used to classify the input image based on its multi-label obtained by the $K$ activation maps. During inference, the activation maps obtained from the self-attention model are used for semantic segmentation.

The work by Xu et al. (2022) proposed a multiclass transformer that learns class tokens in a classification setting. The basic idea is to learn the interaction between the patches and different class tokens and use the resulting similarity to refine the resulting activation maps from the transformer, which is later used for semantic segmentation. The training loss comprises two multi-label soft margin losses. One is based on the class tokens and the other is based on patch tokens. These losses are calculated between the ground-truth labels at the image level and the predictions for each class. Their implementation depends on a pretrained model on ImageNet and they used multi-scale testing (i.e., processing the same input image at different scales and then combining the results) and CRFs for post-processing.

The work by Xie et al. (2022) introduced a class-agnostic contrastive learning framework for semantic segmentation (C2AM) with pretrained models as backbone for features extraction. Using a pretrained model activation maps for extracting background and foreground features, a contrastive learning approach is used to separate foreground and background by encouraging the similarity within each foreground and background representation and dissimilarity across foreground and background representations. The foreground and background representations are separated by a matrix multiplication with a thresholded mask based on the value of the activations. The mask is obtained by the pretrained model's activation maps.

Overall, the work presented in this section relies on pretrained models, typically trained on large datasets. We demonstrate that training models from scratch on synthetic data does not yield useful CAMs, which are crucial for achieving high performance in this line of work.

## A.4 Other Work

Hung et al. (2019) proposed SCOPS, a self-supervised part segmentation framework that leveraged multiple auxiliary objectives. First, they used saliency maps extracted from a pretrained classification network as soft supervision, encouraging the encoder's part activation maps to align with semantically meaningful regions via a cosine similarity loss. Second, they introduced a geometric loss that encouraged all pixels assigned to a given part to be spatially concentrated around that part's center; the number of parts was treated as a hyperparameter. Lastly, an equivariance loss ensured that part assignments remained consistent under random spatial transformations such as rotation and scaling, promoting robustness and part consistency.

The work by Savarese et al. (2021) proposed an unsupervised model-free method to segment foreground and background based on minimizing the mutual information between the segmented foreground and background. They used a variant of mutual information, namely the coefficient of constraint that prevents trivial solution including an entropy in the denominator. By including assumptions about the conditional distributions (i.e., Laplace distribution for foreground/background given background/foreground), the loss is equivalent

to maximizing the inpainting errors of both the masked foreground and the masked background. However, they added a new loss that penalizes the texture diversity within the masked region (i.e., the masked region should have a uniform texture). The mask is estimated for each image by projected gradient descent on the mutual information loss.

## B    Statistical Divergences

In this section, we review statistical divergences, particularly the sliced Wasserstein distance, which serves as the backbone loss for our approach, as explained in the next section.

Previous work in unsupervised foreground object masking (see Section 2) has relied on discriminator networks and adversarial loss functions as in the original GAN (Goodfellow et al., 2014), which exploit the fact that the cross-entropy loss of the ideal discriminator is proportional to the Jensen-Shannon divergence. Since Goodfellow et al. (2014)'s seminal work, a variety of statistical divergences have been used to train GANs, namely $f$-divergences (Nowozin et al., 2016) and maximum mean discrepancy (MMD) (Li et al., 2017).

The family of Wasserstein distances is another class of divergences based on optimal transport theory (Villani, 2008). In particular, the Wasserstein-1 distance, a statistical divergence also known as the Earth mover's distance, has been applied in generative modeling in the form of the Kantorovich dual (Arjovsky et al., 2017). Note that all of these divergences require adversarial optimization (MMD-GAN (Li et al., 2017) uses adversarial selection of the kernel). In contrast, the primal form of the Wasserstein-$p$ distance measures the divergence between the distributions of two random variables $X \sim \mathbb{P}_X$ and $Y \sim \mathbb{P}_Y$ in terms of the optimal transport plan

$$W_p(\mathbb{P}_X, \mathbb{P}_Y) \triangleq \left( \inf_{\mathbb{P}_{XY} \in \mathcal{P}_{\mathbb{P}_X, \mathbb{P}_Y}} \mathbb{E}_{(X,Y) \sim \mathbb{P}_{XY}} d(X,Y)^p \right)^{\frac{1}{p}}, \tag{10}$$

where $\mathcal{P}_{\mathbb{P}_X, \mathbb{P}_Y}$ defines the set of transport plans, consisting of all joint distributions with marginals equal to the original distributions $\mathbb{P}_X$ and $\mathbb{P}_Y$, and $d$ is a distance function. For $X, Y \in \mathbb{R}^d$, the Euclidean distance is typically used $d(X,Y) = \|X - Y\|_2$.

In practice, only samples are available, and for training divergence estimates require computation on batches. For batches of size $N$, the computational complexity of variational approaches (Nowozin et al., 2016; Arjovsky et al., 2017) is $\mathcal{O}(N)$, compared to $\mathcal{O}(N^2)$ for MMD (Li et al., 2017), and $\mathcal{O}(N^3)$ for the primal form of Wasserstein-$p$ distance. To mitigate the computational bottleneck of the Wasserstein distance for samples, the Sinkhorn algorithm has been applied to solve an entropically regularized version of equation 10 (Cuturi, 2013). Another alternative, is to exploit the sliced Wasserstein distance (Kolouri et al., 2018), as the one-dimensional case of Wasserstein-$p$ distance requires only $\mathcal{O}(N \log N)$, due to a closed form (Santambrogio, 2015) that requires sorting. The sliced Wasserstein (SW) distance is motivated by the Cramér-Wold theorem (Bélisle et al., 1997), which states that probability measures in Euclidean space are characterized by all one-dimensional projections, which corresponds to the integration of the Wasserstein distance across all one-dimensional slices (Wu et al., 2019; Deshpande et al., 2018; Kolouri et al., 2018). In practice, this integration is an expectation that can be computed by Monte Carlo integration by sampling from a uniform distribution over the unit hypersphere $\mathbb{S}^{d-1}$.

$$\begin{aligned} \overline{\mathrm{SW}}_p^p(\mathbb{P}_X, \mathbb{P}_Y) &\triangleq \int_{\mathbb{S}^{d-1}} \mathrm{SW}_p^p(\mathbb{P}_X, \mathbb{P}_Y; w) \, dw \\ &= \mathbb{E}_{W \sim U(\mathbb{S}^{d-1})} \left[ \mathrm{SW}_p^p(\mathbb{P}_X, \mathbb{P}_Y; W) \right], \\ \mathrm{SW}_p(\mathbb{P}_X, \mathbb{P}_Y; w) &\triangleq W_p(\mathbb{P}_{w^\top X}, \mathbb{P}_{w^\top Y}). \end{aligned} \tag{11}$$

Given two samples/batches $\hat{\mathbb{P}}_X = \{(\frac{1}{N}, \mathbf{x}_i)\}_{i=1}^N$ and $\hat{\mathbb{P}}_Y = \{(\frac{1}{N}, \mathbf{y}_i)\}_{i=1}^N$, the computation of the sliced Wasserstein-$p$ distance is simply the mean of the absolute error to the $p$th power between paired samples after sorting: $\mathrm{SW}_p^p(\hat{\mathbb{P}}_X, \hat{\mathbb{P}}_Y; \mathbf{w}) = \frac{1}{N} \sum_{j=1}^N (\mathbf{w}^\top \mathbf{x}_{\pi_j} - \mathbf{w}^\top \mathbf{y}_{\sigma_j})^p$, where the permutations $\pi$ and $\sigma$ ensure that $\mathbf{w}^\top \mathbf{x}_{\pi_1} \leq \mathbf{w}^\top \mathbf{x}_{\pi_2} \leq \cdots \leq \mathbf{w}^\top \mathbf{x}_{\pi_N}$ and $\mathbf{w}^\top \mathbf{y}_{\sigma_1} \leq \mathbf{w}^\top \mathbf{y}_{\sigma_2} \leq \cdots \leq \mathbf{w}^\top \mathbf{y}_{\sigma_N}$, respectively. Intuitively, $\mathbf{w}$ defines a subspace and the sorting ensures the shortest distances between the pairs in the subspace.

Alternatively, one can use energy-based Wasserstein (EBSW) (Nguyen & Ho, 2023) that depends on importance sampling to emphasize slices that distinguish the distributions:

$$\text{EBSW}_p(\mathbb{P}_X, \mathbb{P}_Y; f) = \left(\mathbb{E}_{W \sim \sigma_{X,Y}(W; f, p)} \left[\text{SW}_p^p(\mathbb{P}_X, \mathbb{P}_Y; W)\right]\right)^{\frac{1}{p}}. \tag{12}$$

With a uniform proposal distribution over the hypersphere,

$$\sigma_{X,Y}(w; f, p) \triangleq \frac{f(\text{SW}_p^p(\mathbb{P}_X, \mathbb{P}_Y; w))}{\int_{\mathbb{S}^{d-1}} f(\text{SW}_p^p(\mathbb{P}_X, \mathbb{P}_Y; w'))dw'}, \tag{13}$$

where $f$ is a monotonic function. Importantly, this avoids an explicit search for the best slice (Deshpande et al., 2019). Using Monte Carlo integration, a set of $L$ random unit vectors $w_1, \ldots, w_L$ are drawn, importance sampling estimate approximate

$$\text{EBSW}_p^p(\mathbb{P}_X, \mathbb{P}_Y; f) \approx \sum_{l=1}^{L} \frac{f(\text{SW}_p^p(\mathbb{P}_X, \mathbb{P}_Y; w_l))}{\sum_{l'=1}^{L} f(\text{SW}_p^p(\mathbb{P}_X, \mathbb{P}_Y; w_{l'}'))} \text{SW}_p^p(\mathbb{P}_X, \mathbb{P}_Y; w_l). \tag{14}$$

Interestingly, an analogous approach can be used to select from a predefined set of kernel functions for MMD (Biggs et al., 2023), which provides a rough approximation of the adversarial optimization of the kernel function in MMD-GAN (Li et al., 2017).

## C  Synthetic Datasets

### C.1  Synthetic Data Description

#### C.1.1  Synthetic-Aperture Sonar Backgrounds with dSprites (SAS+dSprites)

This synthetic dataset is derived from real-world background-only synthetic-aperture sonar (SAS) images (Cobb & Zare, 2014), augmented by overlaying artificial objects to simulate relevant scenarios. This SAS dataset contains 129 $1001 \times 1001$ images that are originally complex-valued, single (high-frequency) channel, taken from various seafloor texture types. For each sonar scene, we crop 200 $64 \times 64$ images from the magnitude images. We normalize each crop by its mean and then clip any value larger than 16. Half of the resulting crops are used as composite images where we paste objects from dSprites (Matthey et al., 2017) dataset using equation 1. The data is divided into $70\%, 20\%$, and $10\%$ for training, validation, and testing, respectively.

We select the dSprites dataset because the shapes resemble real man-made objects observed in sonar images. However, since the dSprites shapes lack intensity variations, we introduced a smooth gradient to each dSprite object, which mimics the consistent intensity patterns typical in SAS imaging systems. This gradient follows a planar transition with a 45-degree slope and is applied based on the corresponding crop values. The minimum value of the transition is chosen randomly between the 50th and 70th percentiles of the crop values, while the maximum value is randomly selected between the 80th and 100th percentiles of the crop values. This dependence on the background intensity values makes the foreground object brighter in a bright crop and dimmer in a dim one. This makes the segmentation task more challenging since using a simple threshold will not produce a reasonable mask. We use the other half as background images.

#### C.1.2  Textures+MNIST/FashionMNIST/dSprites

This synthetic dataset consists of surrogate data, represented as grayscale single-channel images, which we use to further demonstrate and validate the method's functionality. We sourced background images from crops of 10 different textures obtained from the Brodatz dataset (Brodatz, 1966). For foreground objects, we used MNIST, Fashion MNIST (Xiao et al., 2017), and dSprites (Matthey et al., 2017). These datasets reflect a variety of foreground objects: MNIST are medium extent, thin objects with high pixel values; Fashion MNIST are objects of large extent, a diversity of shapes, some almost-convex, and varying in pixel intensities; and dSprites are small compact shapes (we created an artificial range of intensity values).

For the synthetic dataset, $140,000$ background images of size $64 \times 64$ were generated by randomly patching the texture images. Half of the background images were used as background for the composite images. This resulted in $70,000$ images as foreground and background. Images in the foreground datasets were resized to $64 \times 64$ and the true (binary) mask was used to blend it onto the background image using equation 1. Since MNIST and Fashion MNIST have no given true mask, we estimate the true mask by applying Otsu's threshold on each image (Otsu, 1979). The data is divided into $70\%, 20\%,$ and $10\%$ for training, validation, and testing, respectively.

Again, since the dSprites images lack intensity variation, we create random intensities for each foreground object pixel by sampling a Gaussian distribution with zero mean and unit standard deviation and then squaring it to remove negative values. We then min-max normalize the resulting composite image such that the lowest intensity value is 150 and the max is 255, before masking a background and combining the two. Each image is min-max normalized such that all pixel values are between $[0, 1]$.

## C.2 Implementation Details for Synthetic Datasets

We note here any dataset dependent implementation details.

For SAS+dSprites and Textures+MNIST/FashionMNIST/dSprites, the surrogate data cluster labels for the composite images are computed by processing the corresponding background-only images through both the AE/CLE and Sinkhorn K-means. For the SAS images, we find that AE architecture is not able to reconstruct the images, so we use a symmetrical architecture that is composed of four blocks for the encoder and decoder, and each block is composed of convolution, ReLU, BatchNorm, with residual units: the 64-channel block has 1 and other blocks have 2 residual units each. The number of channels per block is $64, 128, 256, 512,$ with an additional 768-channel stage. As described above, a fully connected bottleneck layer of dimension $o = 20$ is used to represent the latent space.

The Textures+MNIST/FashionMNIST/dSprites images, have 10 ground truth clusters and we experiment with varying the number of clusters, choosing values from the set $\{5, 10, 15\}$. For SAS+dSprites images, we choose cluster values from the set $\{3, 4, 5\}$. Initial experiments with 2 clusters consistently failed to separate visually distinct regions.

For Textures+MNIST/FashionMNIST/dSprites and SAS+dSprites, we run the training for 50 and 100 epochs, respectively. For the former, we choose the model with lowest validation loss $\mathcal{L}$ shown in equation 8, but for the latter we also include a quantile loss equation 15 with quantile, $q = 0.8$, such that the maximum area of an object is $20\%$ of the image size:

$$\mathcal{L}_{\text{quantile}}(\theta) = \mathbb{E}_{X,R} \left[ \sum_{ij \in \mathcal{Q}} \left( [\tilde{X}_\theta]_{ij} - R_{ij} \right)^2 \right], \quad \mathcal{Q} = \{ij : M_\theta(X)_{ij} \leq \tau\}, \quad \tau = \text{quantile} \left( M_\theta(X), q \right). \quad (15)$$

That is, for the SAS+dSprites dataset the total loss is $\mathcal{L}_D(\theta) + \mathcal{L}_{\text{Bg}}(\theta) + \mathcal{L}_{\text{quantile}}(\theta)$. This additional loss encourages the masking network to produce counterfactual images in which a $q$th fraction of its pixels match the source background, essentially encouraging the corresponding mask pixels to be zero.

## C.3 Additional Results for Textures+MNIST/FashionMNIST/dSprites

Figure 10 shows the clustered background using AE+K-means along with the corresponding true clusters.

Figure 11 shows qualitative results for the masks using the true texture classes.

## C.4 Unsupervised Object Segmentation Benchmarks for Textures+MNIST/FashionMNIST/dSprites

For the synthetic datasets, since we have more class information about the background and objects, we can train a classification model and use its resulting CAM. We use the method introduced by Xie et al. (2022) as a benchmark as it gives the best results among the CAM-based models. We find it one of the closest methods to our work due to its weakly supervised nature. We also use the GAN-based method, PerturbGAN, introduced

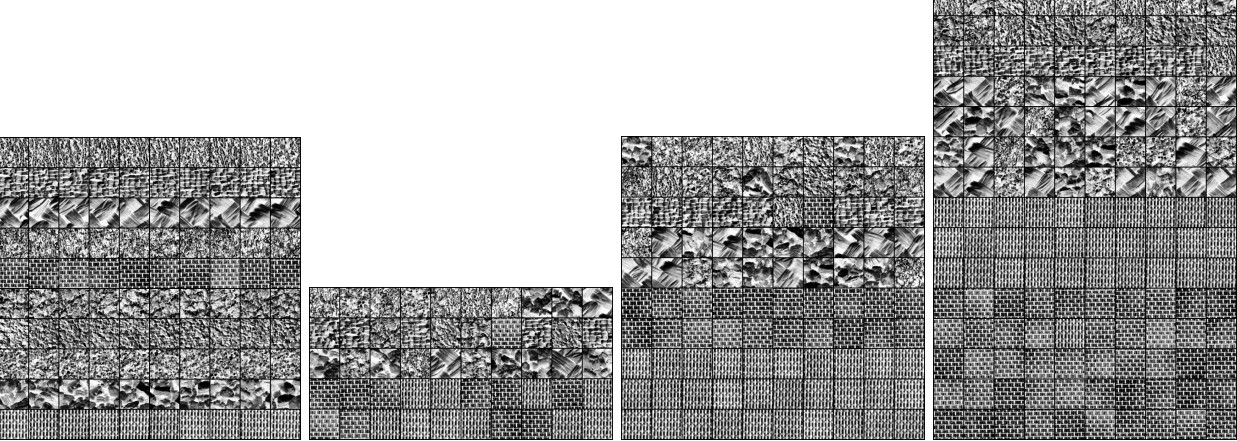

Figure 10: Background Clusters for Textures: From left to right: True clusters, 5, 10, and 15 clusters.

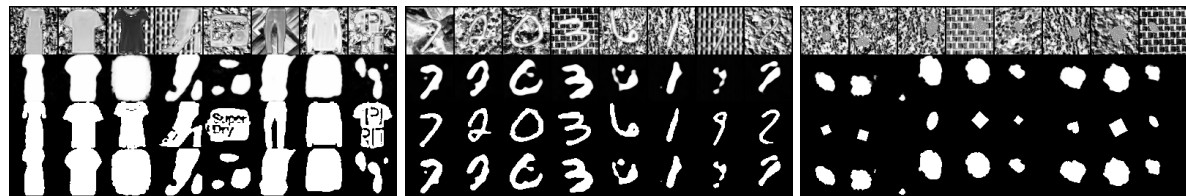

Figure 11: Results of the segmentation model for Texture+Fashion MNIST (left), Texture+MNIST (middle), and Texture+dSprites (right) using true clusters. From top to bottom: input images, estimated masks, true masks, and thresholded estimated masks at 0.5

by Bielski & Favaro (2019). Both of them fail to converge to meaningful object masks when using synthetic single-channel data. In order to compare the performance of the related work using our synthetic data, we train PerturbGAN (Bielski & Favaro, 2019) on our composite images (excluding background images). We also found that the work by Xie et al. (2022) (C2AM) can be adapted to take advantage of the composite and background images. Thus, we train a ResNet-52 model (their backbone model) to classify the composite images based on their foreground objects class, and to classify background images based on their texture label. We choose this classification task to encourage the ResNet model to learn a rich representation that has enough entropy to be used as CAM. After training the backbone model, we proceed to fine-tune the model using their approach.

We have noticed that PerturbGAN is able to generate similar images to the synthetic dataset; however, it was able to 'cheat' as the generated foreground mask did not represent the true foreground masks. Also, the generated background images have composite images and background images as shown in Figure 13.

For the C2AM model (Xie et al., 2022), we notice that it failed to converge to a meaningful solution as shown in Figure 12. Note that these are training samples. We hypothesize this is due to a not well-trained backbone model; the model either needs more data or a more complicated modeling scheme, for which the two scenarios invalidate the minimalist approach that our method promotes.

# D   Additional Results

## D.1   Additional Results for AI4Shipwrecks

Figure 14 shows cases where the masking network also includes shadows of ships.

In contrast, Figure 16 shows cases where the masking network is successful with less extensive shadows.

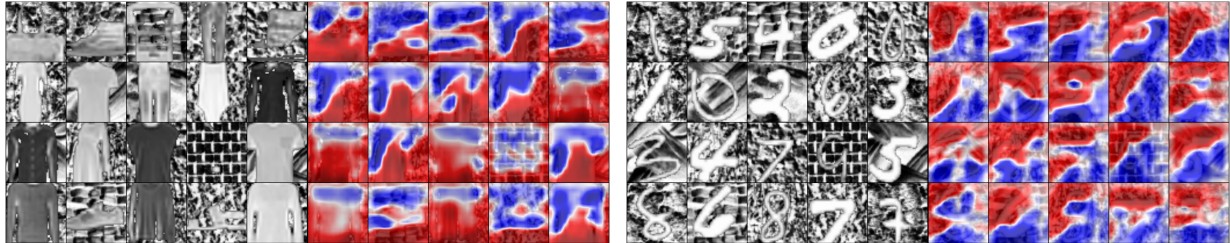

Figure 12: Results from C2AM model based on Fashion MNIST (left) and MNIST (right) datasets. Left of each block: the training input image. Right of each block: the activation maps where red shows the predicted foreground object.

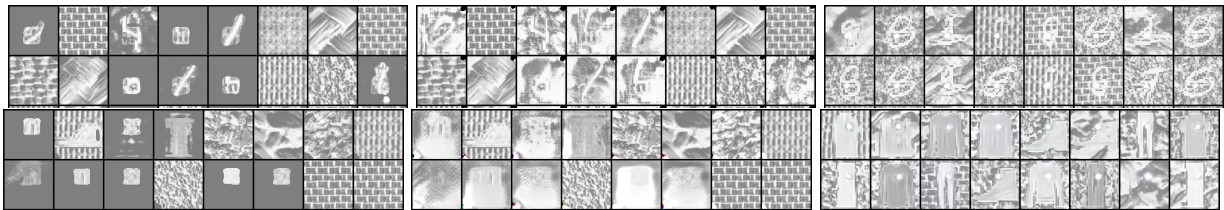

Figure 13: Results from PerturbGAN images based on MNIST (top) and Fashion MNIST (bottom) datasets. From left to right: generated mask along with the generated composite images, generated composite images, and generated background images.

Figure 15 shows images without ships, where the model identifies the artificial reef, although it not in the ground truth. As a man-made object and arrangement it could be argued that the masking network worked in this case. However, Figure 15 also highlights an image where there are extensive false positives for portions of the seafloor.

Figure 17 is another case with many false positives, that other supervised models also perform poorly on.

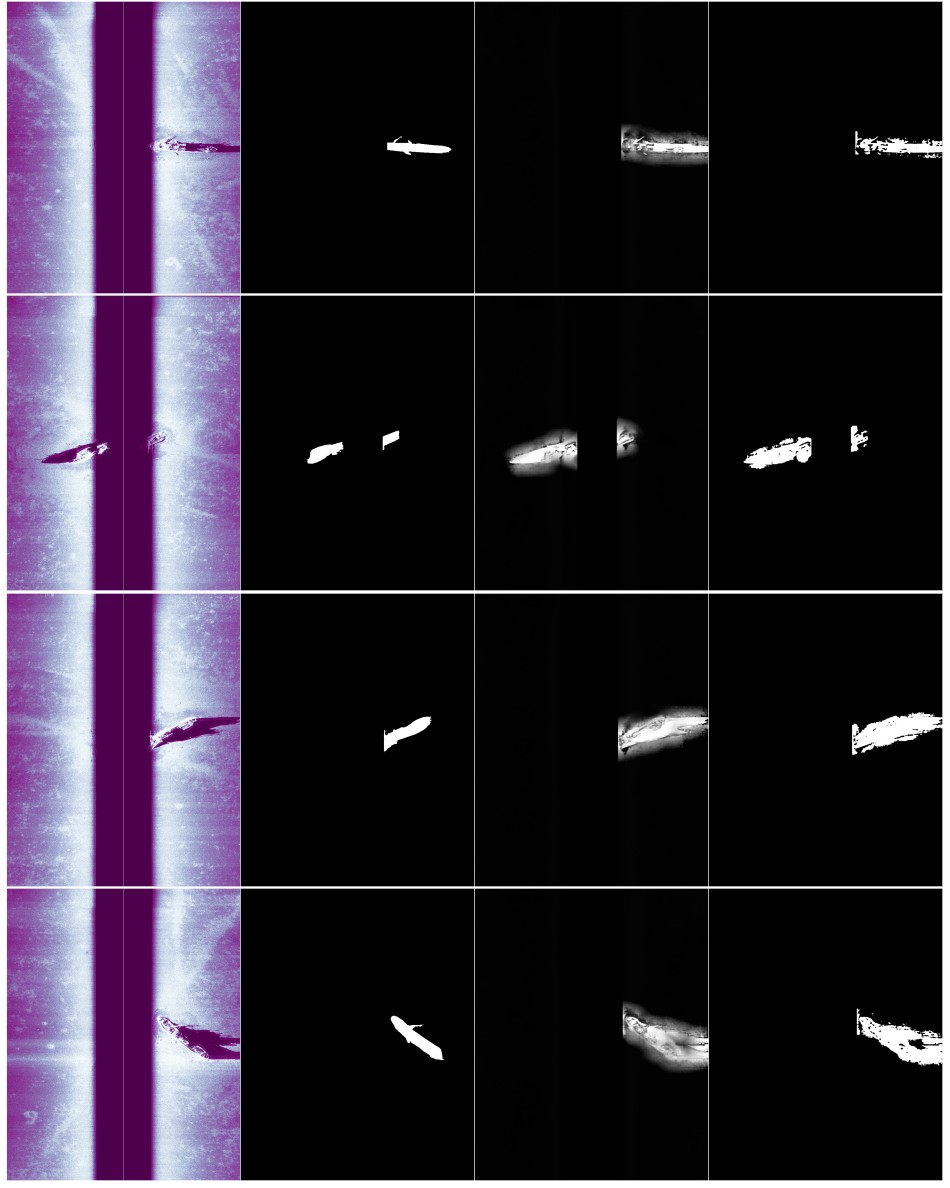

Figure 14: Segmentation model results on AI4Shipwrecks. Each row corresponds to a different image. From left to right: input images, ground truth masks, estimated masks, and thresholded estimated masks at 0.5. Image sites: Viator.

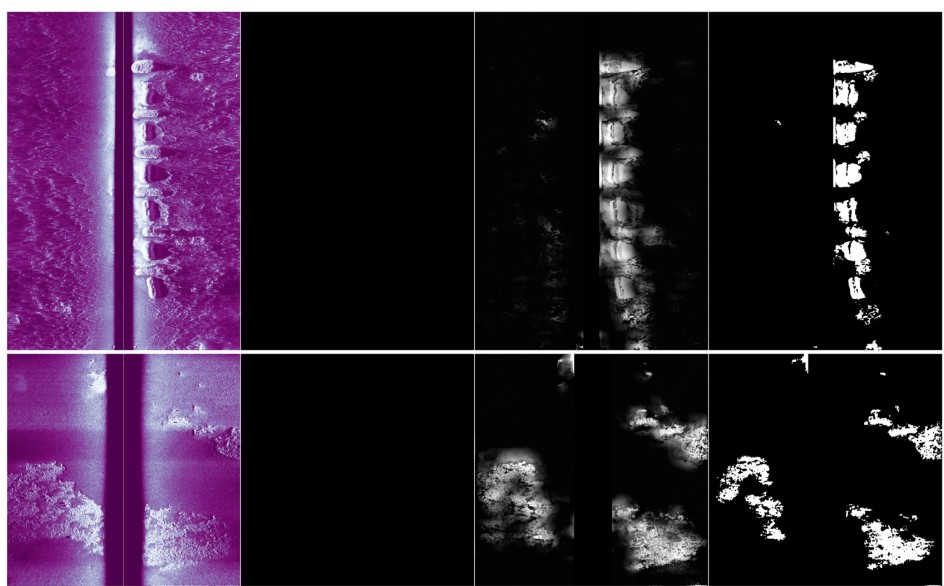

Figure 15: Segmentation model results on images without ships from AI4Shipwrecks, but regular arrangement of rocks or blocks in the first image. Each row corresponds to a different image. From left to right: input images, ground truth masks, estimated masks, and thresholded estimated masks at 0.5. Site of the first image is Artificial Reef, and for the second image is Haltiner Berge.

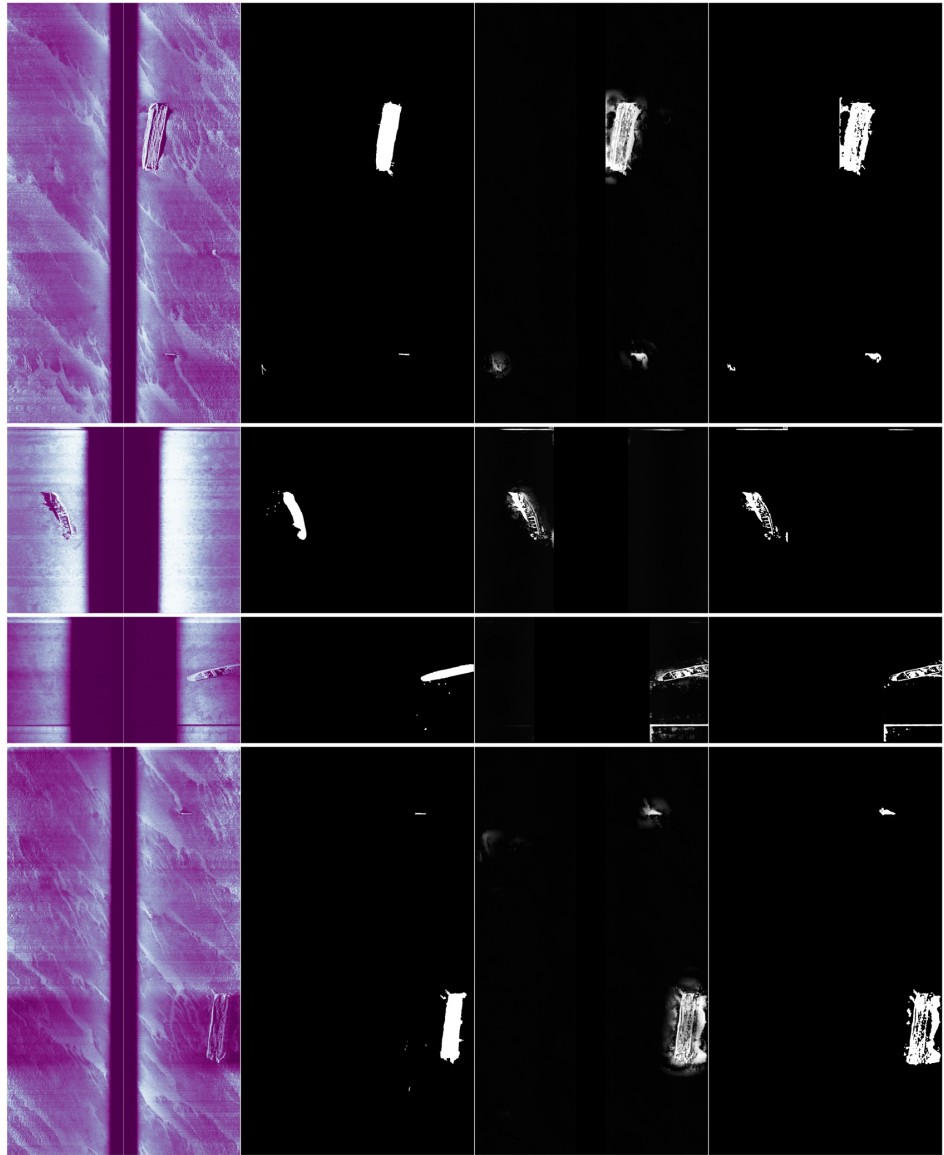

Figure 16: Segmentation model results on AI4Shipwrecks. Each row corresponds to a different image. From left to right: input images, ground truth masks, estimated masks, and thresholded estimated masks at 0.5. Images sites from top to bottom: Barge, WH Gilbert , WH Gilbert, and Barge.

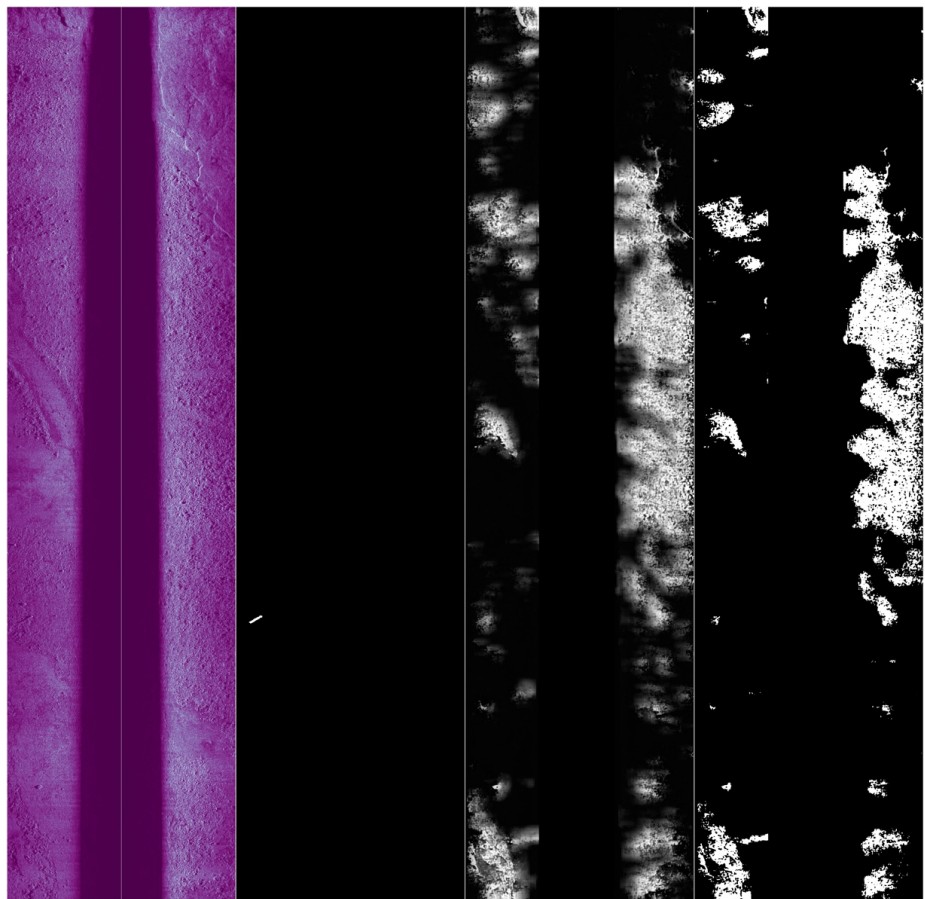

Figure 17: Failure cases of the segmentation model on images at 'James Davidson' from AI4Shipwrecks. From left to right: input image, ground truth mask, estimated mask, and thresholded estimated mask at 0.5.

### D.1.1 Unsupervised Benchmark Results on AI4Shipwrecks

Attempts to apply unsupervised segmentation methods to the AI4Shipwrecks dataset are shown in Figures 18, 19, 20, and 21.

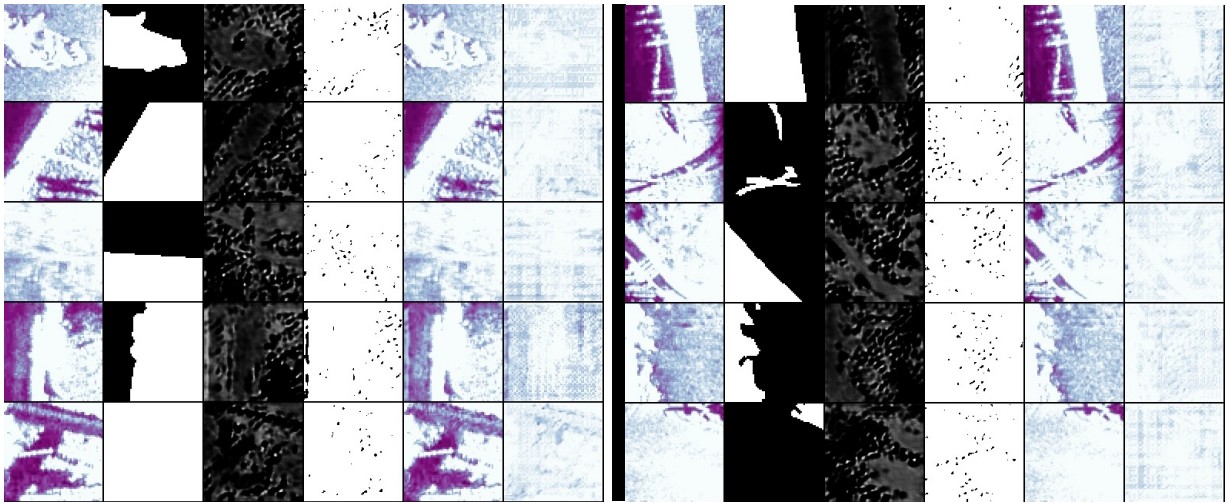

Figure 18: Two blocks of results from Chen et al. (2019) model for AI4Shipwrecks. Each row represents the result of an input sonar patch: first column is the input sonar patch, second is the segmentation ground truth, third and fourth are generated masks, and the last two are alpha blended results based on the generated masks.

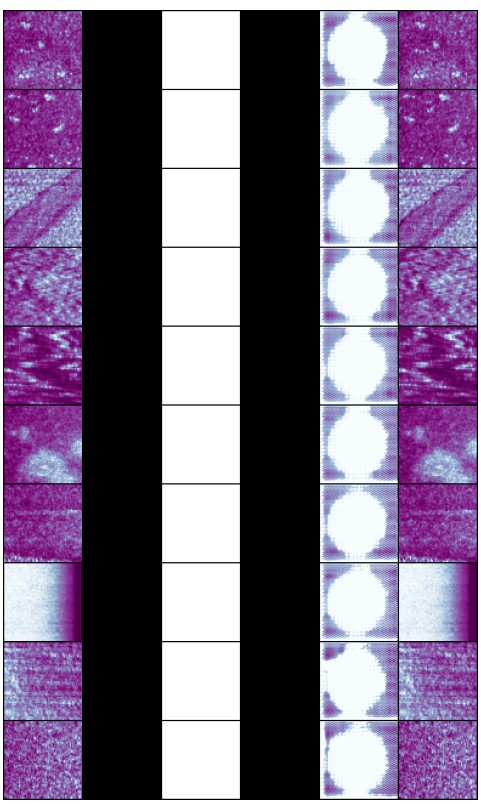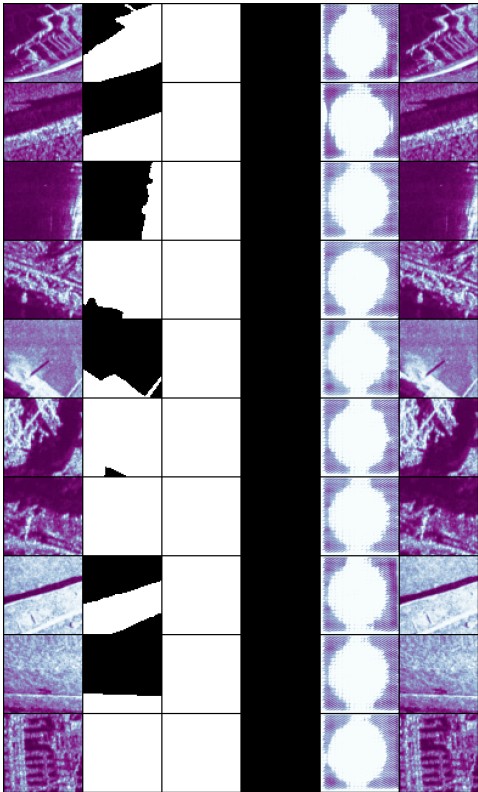

Figure 19: Two blocks of results from Chen et al. (2019) model for AI4Shipwrecks. Each row represents the result of an input sonar patch: first column is the input sonar patch, second is the segmentation ground truth, third and fourth are generated masks, and the last two are alpha blended results based on the generated masks. The left figure shows a group of background patches and the right one shows a group of composite images.

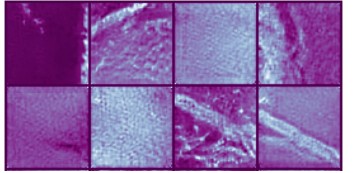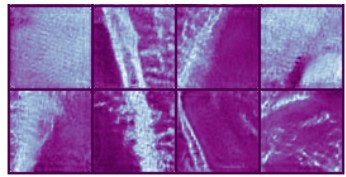

Figure 20: Generated images from ILSGAN (Zou et al., 2023) for AI4Shipwrecks. Each cell represents a generated image.

.

Figure 21: Results from model-free unsupervised baseline method Savarese et al. (2021) using patches with 50% overlaps on AI4Shipwrecks.

### D.2 Additional Results for CUB Dataset

Figure 22 shows examples of the background images extracted from original CUB split organized into 10 clusters. Note that color and texture are very distinct.

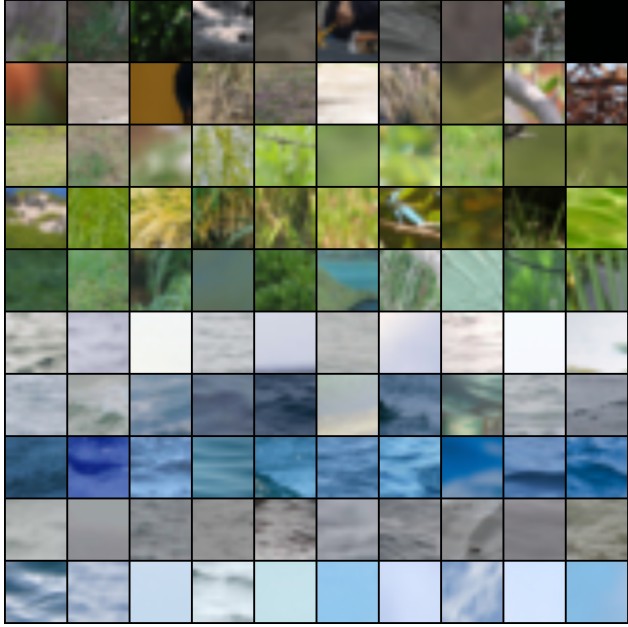

Figure 22: Each row shows an example obtained from clusters for the CUB dataset.

For quantitative results, we show a sensitivity analysis of performance metrics for the choice of latent representation (auto-encoder or contrastive learning) and the number of clusters for the original CUB split are shown in Table 7.

Table 7: Masking Network Performance (Average Precision, AUCROC and IoU) Comparison of AE+K-means and CL+K-Means based Background Clustering

| Metric | Number of Clusters | AE | CL |
|---|---|---|---|
| **Average Precision** | 5 | 34% | **46%** |
| | 10 | 38% | **40%** |
| | 15 | 37% | **41%** |
| **AUCROC** | 5 | 72% | **74%** |
| | 10 | 67% | **70%** |
| | 15 | 70% | **74%** |
| **IoU** | 5 | 20% | **30%** |
| | 10 | 24% | **30%** |
| | 15 | 24% | **26%** |

