# OpenReview forum: "Weakly Supervised Object Segmentation by Background Conditional Divergence"
_TMLR — Accepted by TMLR_

### Review · Reviewer_RDqq · 2025-07-23

**Summary Of Contributions:**

This paper proposes a method for weakly supervised binary segmentation.
The proposed approach trains a model to predict a mask that separates an image into foreground and background regions.
The model is trained using foreground images with masks and background images, by minimizing the divergence between outputs on counterfactual images generated from them.

**Audience:**

No

**Broader Impact Concerns:**

There appear to be no significant ethical concerns related to this work.

**Claims And Evidence:**

No

**Requested Changes:**

# Major Concerns
- Clarifying the effectives of the proposed method compared to copy-and-paste based approaches [1][2][3].
- Provide a more detailed discussion of Table 3.

[1] Cut, Paste and Learn: Surprisingly Easy Synthesis for Instance Detection, ICCV 2017
[2] Modeling Visual Context is Key to Augmenting Object Detection Datasets, ECCV 2018
[3] InstaBoost: Boosting Instance Segmentation via Probability Map Guided Copy-Pasting, ICCV 2019

**Strengths And Weaknesses:**

# Pros
- This paper demonstrates the performance of the proposed model on several benchmark datasets.
- The idea behind the proposed method is reasonable and has the potential to outperform GAN-based approaches.

# Cons
- This paper is missing several related works [1][2][3], which consider similar setups where both foreground and background images are provided for training a segmentation model.
- It is unclear why such a complex training scheme is necessary. In my opinion, synthesizing training data through simple copy-and-paste may be sufficient for training the model.
- The discussion of Table 3 is limited. It is not clear why the proposed method achieves lower scores compared to the baseline methods.

[1] Cut, Paste and Learn: Surprisingly Easy Synthesis for Instance Detection, ICCV 2017
[2] Modeling Visual Context is Key to Augmenting Object Detection Datasets, ECCV 2018
[3] InstaBoost: Boosting Instance Segmentation via Probability Map Guided Copy-Pasting, ICCV 2019

---

> ### Author Response · Authors · 2025-09-03
> **Addressing concerns and questions.**
>
> We thank the reviewer for their time. Based on the comments, we have made some revisions and you can see it in the revised version where the new additions are colored with blue and parts moved from the appendix are colored with maroon.
>
> Regarding the first request: “This paper is missing several related works [1][2][3], which consider similar setups where both foreground and background images are provided for training a segmentation model.” We note that the references are all related to creating synthetic images (of the same form as our counterfactual images) using known ground truth masks for the purpose of training subsequent instance segmentation or object detectors. This previous work is motivated by augmenting the classification dataset to have a higher grained classification or to boost the performance. We note that the end goals of three references concur with our end-goal, which is to improve performance on real data, but the known masks used to create more  synthetic supervision are not compatible with our weakly supervised setting.
>
> The “Cut, Paste and Learn” method (Dwibedi et al., 2017) assumes access to images on “modest backgrounds” with depth maps. The mask network is then built from a pre-trained model with the depth as ground truth. The masked objects are then placed on various scenes and different blending (Gaussian blurring and solving the Poisson equation for Poisson image editing) is applied to minimize local artifacts before training the detection methods on these synthetic images with known masks. In contrast, our approach assumes coarse detection (for the weak supervision) but then learns to mask from objects in the wild. That is, our method follows almost an opposite learning path. One insight is that differentiable blending options could be applied after the alpha-blending to help minimize the divergence between the counterfactual and real images.
>
> The paper by Dvornick et al. (2018) is focused on creating synthetic images using objects with ground truth masks by placing them in appropriate locations in background images, a task noted in our Footnote 1, which mentions the Topnet work by Zhu et al. (2023). (We will add the reference to Dvornick et al. there.) Like “Cut, Paste and Learn” the method pastes objects on background images as an augmentation technique, but places the objects based on surrounding visual cues on the background images. The idea is that random placement can deteriorate performance of the downstream task. As noted in our Footnote 1, “We assume the object can be placed on an independently chosen background. This assumption is valid in underwater sonar where a man-made object’s location is largely independent of the background.” We have added a Footnote 2 to also note this in the methodology.
>
> The InstaBoost paper by Fang et al. (2019) follows along the same lines as the two other references by requiring true masks for objects and synthesizing new images to train subsequent semantic segmentation. Like Dvornick et al. (2018) it seeks optimal placement, but it does so within the same image, rather than a different image, using a “consistency” heatmap based on a distance defined between the original background surrounding the object to the background surrounding it in potential locations. Additionally,  because it is the same image, the approach needs to inpaint the original location when pasting to a new location. Thus, while the idea of creating counterfactuals within the same image could be explored, it is not clear that the near-object background pixels distance would work in other imaging domains, especially the grayscale intensity images in SONAR.
>
> In contrast, we do not assume any knowledge about the masks on the objects: we only need composite and background images.  If we had ground truth masks for objects we could certainly incorporate them using the “Cut, Paste and Learn” approach—incorporating the blending and augmentations, as additional synthetic supervision to be combined with weakly supervised real cases. We have now added this potential to the Discussion (section 5).
>
> We have moved the summary of results of the synthetic dataset to the main body now (section 4.3 and 4.4). We also added a table of notation (new Table 1) and an algorithm summarizing the training procedure (Algorithm 1). We also added two paragraphs in section 4.2.1 to discuss the CUB datasets results. Finally, we added a conclusion (section 6) to summarize the paper.
>
> Regarding the performance in Table 3 (now Table 4), performance on CUB is modest due to a number of issues that were noted in response to Reviewers jGih. A statement of this discussion is now included following the table in section 4.2.1.

---

### Review · Reviewer_ARY3 · 2025-07-24

**Summary Of Contributions:**

The paper proposes a new method for weakly supervised object segmentation, where the only labels available during training are binary labels indicating if the object is present in the image or not. First, the proposed method clusters the background-only images relying on the features extracted by either a VAE or an encoder trained via contrastive learning. Then, during training, for each image the segmented object is placed on a background from a different cluster, yielding counterfactual images. The training objective aims to minimize the divergence between counterfactual images and original images from the target cluster. Since this approach does not require expensive pre-training, ground-truth segmentation masks or complex training pipelines, it is well-suited for domains where collecting and labelling data is expensive or not possible. In the experimental evaluation, the proposed approach is competitive with supervised methods on sonar images datasets, where unsupervised methods fail. Moreover, it achieves non-trivial results on a natural images task.

**Audience:**

Yes

**Broader Impact Concerns:**

No concern.

**Claims And Evidence:**

Yes

**Requested Changes:**

- I think the results in Table 3 and limitations on CUB should be better discussed in Sec. 4.2.1.

- The paper mentions several times the toy (synthetic) datasets, but the results of the corresponding experiments are not discussed in the main part. While the results with real images are more interesting, I think a short summary of the results on the toy datasets should be added.

- As minor point, testing the proposed method on another tasks with limited data (not necessarily natural images) besides the sonar images datasets would strengthen the paper.

**Strengths And Weaknesses:**

Strengths
- The proposed approach is well motivated and presented. It needs only labels on the presence of the object in the image, which are significantly less expensive to collect than the full segmentation masks, and no pre-training, which makes it an interesting option for tasks with limited data.

- The experiments support the effectiveness of the proposed method on datasets of sonar images.

Weaknesses
- The proposed approach doesn't appear competitive with existing (unsupervised) methods on the CUB dataset (see Table 3). As acknowledged in the text, it is not clear how well the proposed method can work when the objects and background distributions are not independent. This questions the applicability to natural images tasks.

---

> ### Author Response · Authors · 2025-09-03
> **Addressing Questions and Changes**
>
> Thank you for your thoughtful review. Based on the comments we have revised the manuscript, with new additions colored blue and parts moved from the appendix are colored maroon.
>
> Regarding the CUB dataset, we have added a discussion in the revised paper (the last two paragraphs in section 4.2.1) and we also discussed in the comments to Reviewer jGih.
>
> We have moved the summary of the results of the synthetic dataset to the main body now (section 4.3 and 4.4). We also added a table of notation (Table 1) and an algorithm summarizing the training procedure (Algorithm 1).  Finally, we added a conclusion (section 6) to summarize the paper.

---

### Review · Reviewer_jGih · 2025-08-18

**Summary Of Contributions:**

The paper proposes a weakly supervised object segmentation method that leverage images with objects in the foreground and background only images. The method trains a masking segmentation network by minimizing the divergence between background-conditioned counterfactual images (via alpha-blend) and real composite images. The background-only images are categorized into different classes to enable the model learn the mask independent of backgrounds. The model can be trained without pixel-level label for foreground and background annotation.

**Audience:**

Yes

**Claims And Evidence:**

Yes

**Requested Changes:**

Please improve the writing and explain the results in the above questions.

**Strengths And Weaknesses:**

Strengths:

- The paper is well-motivated to solve segmentation problems in specific domain with few annotations.

- The proposed method is reasonable in the setting to enable WSSS via only binary labels of whether the image contains the object in the foreground.

Weaknesses:

- The writing of the paper is hard to follow. For example, Sec 3.1 problem formulation does not clear state what problem to solve, while it seems to explain the whole method and loss function, which is hard to follow.

- The notations in the paper are not rigorously defined. For example, in Eq 4, what is $M_{degen}$? What is X->1? What is P_X actually is? There is no probability properly defined here.

-  Given the unorganized structure and notations, it is hard to reproduce the methodology after only reading the paper. Instead of vaguely writing formulas of expectations and divergence, I recommend authors writing down concrete algorithms on how are data sampled, network input/outputs and what objective function given sampled data.

- The paper only compared with supervised segmentation methods and admits that it does not work as good as supervised methods, but the comparison to other WSSS method lacks. Also, the experiments results on toy dataset is missing and the performance on the CUB dataset does not looks promising.

---

> ### Author Response · Authors · 2025-09-03
> **Addressing Weanesses and Questions**
>
> Thank you for your review, which has led to an improved manuscript, and for recognizing the motivation of the problem in data scarce regimes. We have uploaded a revised version where new additions are colored with blue and parts moved from the appendix are colored maroon.
>
> To address the first three weaknesses we added Table 1 to summarize the symbols and Algorithm 1 to summarize the training procedure for the masking network. We have clarified a few points regarding the distributions and defined empirical distributions. As stated in the new table, and revised text, $M_\text{degen}: X \mapsto \mathbf{1},\quad \forall X\in\mathcal{X}$. The losses in the algorithm are defined in terms of sampled data through the empirical distributions.
>
> Regarding comparisons beyond supervised segmentation, we had conducted a thorough comparative analysis against unsupervised and weakly supervised state-of-the-art approaches. To our knowledge, there are no existing weakly-supervised semantic segmentation (WSSS) methods specifically designed for sonar images.  As detailed in section 4.1.2, we adapted and evaluated three representative unsupervised or weakly supervised methods:
>
> 1. ILSGAN: A GAN-based SOTA method known for its performance on benchmark datasets like CUB.
> 2. C2AM: A top-performing CAM-based model.
> 3. A training-free method (Savarse et al. 2021): A promising, albeit different, approach that we included for its unique methodology.
>
>
> The results, presented in Appendix D.2, show that none of these methods were able to produce meaningful segmentations on our sonar dataset. This highlights the unique challenges posed by sonar imagery and validates the need for our proposed method.
> In appendix C.2.1, we showed that two of the related works (PerturbedGAN which is based on layered GAN and C2AM which is based on CAM— a weakly supervised method) fail to capture the foreground objects when using our synthetic/toy dataset, which are arguably easier but  similar to the sonar data.
>
> The synthetic/toy data was in the appendix but we moved it to the main body (section 4.3 and 4.4). We have moved a summary of the results of the synthetic dataset to the main body. We also added a conclusion (section 6) to summarize the paper.
>
>
> We added two paragraphs in section 4.2.1 to discuss the CUB datasets results as follows:   Our model's modest performance is  characteristic of its objective: generating plausible counterfactuals. The masking network finds a solution not aligned to the ground truth due to three main reasons: it can create plausible counterfactuals by taking portions of images that create bird-like silhouettes, as seen in Figure 5 rows 4, 6, 9, 10 and 11; it can ignore portions of birds bodies that can vary in color, such that when blended with background they are still reasonable representations (perhaps of different species); and thirdly, the strong contextual dependency between foreground objects and their backgrounds. When a strong dependency exists, a cleanly segmented foreground object often creates a statistically improbable image when placed in a new context, resulting in a high loss. Instead of this, our model strategically finds masks, which often include ‘contextual anchors’ from the source background to preserve local realism and create a more plausible final image, as seen in Figure 4. For instance, consider a task where the source image is a bird native to a dense forest, and the target distribution is characterized by backgrounds that are mostly open sky with a few branches. To minimize the statistical distance, the model will not just segment the bird; it will segment the 'bird-on-branch' unit together. A forest bird floating against a plain sky is a statistical anomaly, but by including the branch, the model preserves a high-probability visual pattern. Similarly, for the same target distribution, creating a silhouette of a flying bird out of any composite image will create a plausible counterfactual with the open sky background.
>
> Together, these masks create counterfactuals that better align with the target distribution, thereby achieving a lower loss, but do not align with the ground truth mask of the bird alone. While this masking behavior leads to imperfect segmentation on CUB, it highlights our method's key strength—it is ideally suited for domains where the foreground and background are decoupled. For example, in sonar imagery, man-made objects like shipwrecks (artificial reefs, abandoned crab/lobster traps, or unexploded ordinances or mines) do not correlate with the surrounding seabed. In this setup, where contextual priors are not a factor, our method excels, proving competitive even with supervised approaches.

---

### Decision · Action_Editor_1Rfy · 2025-10-06

**Recommendation:** Accept as is

**Additional Comments:**

In the final recommendation, Reviewer jGih argues that "the experiment and discussion of the results should be improved". However, the reviewer provides no specific comments on what improvements they would like to see. The AE believes that the revisions done by the authors during the discussion addressed the main concerns of the reviewer in that respect and are thus sufficient.

This being said, the main paper is now fairly long, and it may be worth for the authors to consider shortening it. The AE nonetheless understands that the additional content was added based on requests from the reviewers.

**Audience:**

Yes

**Audience Explanation:**

All the reviewers acknowledge that there is an audience for this work.

**Claims And Evidence:**

Yes

**Claims Explanation:**

Two reviewers acknowledge that the claims are sufficiently supported while one does not. The main concern, shared by all reviewers, is the effectiveness of the proposed weakly-supervised strategy in realistic situations. The authors convincingly showcase the effectiveness of their method on Sonar data, and now provide a rationale of why it would not work in some other situations such as on the CUB dataset. The AE agrees with the reviewers that this is a borderline situation; however, there is evidence that the method is useful in some practical cases.